METHODS

# When predict can also explain: Few-shot prediction to select better neural latents

Kabir V. Dabholkar[1]*, Omri Barak[2]

1 Faculty of Mathematics, Technion – Israel Institute of Technology, Haifa, Israel, 2 Rappaport Faculty of Medicine and Network Biology Research Laboratory, Technion – Israel Institute of Technology, Haifa, Israel

* kabir@campus.technion.ac.il

## Abstract

Latent variable models serve as powerful tools to infer underlying dynamics from observed neural activity. Ideally, the inferred dynamics should align with true ones. However, due to the absence of ground truth data, prediction benchmarks are often employed as proxies. One widely-used method, *co-smoothing*, involves jointly estimating latent variables and predicting observations along held-out channels to assess model performance. In this study, we reveal the limitations of the co-smoothing prediction framework and propose a remedy. Using a student-teacher setup, we demonstrate that models with high co-smoothing can have arbitrary extraneous dynamics in their latent representations. To address this, we introduce a secondary metric—*few-shot co-smoothing*, performing regression from the latent variables to held-out neurons in the data using fewer trials. Our results indicate that among models with near-optimal co-smoothing, those with extraneous dynamics underperform in the few-shot co-smoothing compared to 'minimal' models that are devoid of such dynamics. We provide analytical insights into the origin of this phenomenon and further validate our findings on four standard neural datasets using a state-of-the-art method: STNDT. In the absence of ground truth, we suggest a novel measure to validate our approach. By cross-decoding the latent variables of all model pairs with high co-smoothing, we identify models with minimal extraneous dynamics. We find a correlation between few-shot co-smoothing performance and this new measure. In summary, we present a novel prediction metric designed to yield latent variables that more accurately reflect the ground truth, offering a significant improvement for latent dynamics inference.

## Author summary

The availability of large scale neural recordings encourages the development of methods to fit models to data. How do we know that the fitted models are loyal to the true underlying dynamics of the brain? A common approach is to use

**Data availability statement:** Code for the few-shot evaluation is available at https://github.com/KabirDabholkar/nlb_tools_fewshot. Code for the HMM simulations is available at https://github.com/KabirDabholkar/hmm_analysis.

**Funding:** This work was supported by the Israel Science Foundation (grant No. 1442/21 to OB) and Human Frontiers Science Program (HFSP) research grant (RGP0017/2021 to OB). The funders had no role in study design, data collection and analysis, decision to publish, or preparation of the manuscript.

**Competing interests:** The authors have declared that no competing interests exist.

prediction scores that use one part of the available data to predict another part. The advantage of predictive scores is that they are general: a wide variety of modelling methods can be evaluated and compared against each other. But does a good predictive score guarantee that we capture the true dynamics in the model? We investigate this by generating synthetic neural data from one model, fitting another model to it, ensuring a high predictive score, and then checking if the two are similar. The result: only partially. We find that the high scoring models always contain the truth, but may also contain additional 'made-up' features. We remedy this issue with a secondary score that tests the model's generalisation to another set of neurons with just a few examples. We demonstrate its applicability with synthetic and real neural data.

## Introduction

In neuroscience, we often have access to simultaneously recorded neurons during certain behaviors. These observations, denoted $X$, offer a window onto the actual hidden (or latent) dynamics of the relevant brain circuit, denoted $Z$ [1]. Although, in general, these dynamics can be complex and high-dimensional, capturing them in a concrete mathematical model opens doors to reverse-engineering, revealing simpler explanations and insights [2,3]. Inferring a model of the $Z$ variables, $\hat{Z}$, also known as latent variable modeling (LVM), is part of the larger field of system identification with applications in many areas outside of neuroscience, such as fluid dynamics [4] and finance [5].

Because we don't have ground truth for $Z$, prediction metrics on held-out parts of $X$ are commonly used as a proxy [6]. However, it has been noted that prediction and explanation are often distinct endeavors [7]. For instance, [8] use an example where ground truth is available to show how different models that all achieve good prediction nevertheless have varied latents that can differ from the ground truth. Such behavior might be expected when using highly expressive models with large latent spaces. Bad prediction with good latents is demonstrated by [9] for the case of chaotic dynamics.

Various regularisation methods on the latents have been suggested to improve the similarity of $Z$ to the ground truth, such as recurrence and priors on external inputs [10], low-dimensionality of trajectories [11], low-rank connectivity [12,13], injectivity constraints from latent to predictions [8], low-tangling [14], and piecewise-linear dynamics [15]. However, the field lacks a quantitative, *prediction-based* metric that credits the simplicity of the latent representation—an aspect essential for interpretability and ultimately scientific discovery, while still enabling comparisons across a wide range of LVM architectures.

Here, we characterise the diversity of model latents achieving high *co-smoothing*, a standard prediction-based framework for Neural LVMs, and demonstrate potential pitfalls of this framework (see Methods for a glossary of terms). We propose a few-shot variant of co-smoothing which, when used in conjunction with co-smoothing,

differentiates varying latents. We verify this approach both on synthetic data settings and a state-of-the-art method on neural data, providing an analytical explanation of why it works in simple settings.

## Results

### Co-smoothing: A cross-validation framework

Let $\boldsymbol{X} \in \mathbb{Z}_{\geq 0}^{T \times N}$ be spiking neural activity of $N$ channels recorded over a finite window of time, i.e., a *trial*, and subsequently quantised into $T$ time-bins. $X_{t,n}$ represents the number of spikes in channel $n$ during time-bin $t$. The dataset $\mathcal{X} := \{\boldsymbol{X}^{(i)}\}_{i=1}^{S}$, partitioned as $\mathcal{X}^{\text{train}}$ and $\mathcal{X}^{\text{test}}$, consists of $S$ trials of the experiment. The latent-variable model (LVM) approach posits that each time-point in the data $\boldsymbol{x}_t^{(i)}$ is a noisy measurement of a latent state $\boldsymbol{z}_t^{(i)}$.

To infer the latent trajectory $\boldsymbol{Z}$ is to learn a mapping $f : \boldsymbol{X} \mapsto \hat{\boldsymbol{Z}}$. On what basis do we validate the inferred $\hat{\boldsymbol{Z}}$? We cannot access the ground truth $\boldsymbol{Z}$, so instead we test the ability of $\hat{\boldsymbol{Z}}$ to predict unseen or held-out data. Data may be held-out in time, e.g., predicting future data points from the past, or in space, e.g., predicting neural activities of one set of neurons (or channels) based on those of another set. The latter is called co-smoothing [6].

The set of $N$ available channels is partitioned into two: $N^{\text{in}}$ held-in channels and $N^{\text{out}}$ held-out channels. The $S$ trials are partitioned into train and test. During training, both channel partitions are available to the model and during test, only the held-in partition is available. During evaluation, the model must generate the $T \times N^{\text{out}}$ rate-predictions $R_{:,\text{out}}$ for the held-out partition. This framework is visualised in Fig 1A.

Importantly, the encoding-step or inference of the latents is done using a full time-window, i.e., analogous to *smoothing* in control-theoretic literature, whereas the decoding step, mapping the latents to predictions of the data is done on individual time-steps:

$$\hat{\boldsymbol{z}}_t = f(\boldsymbol{X}_{:,\text{in}}; t) \tag{1}$$

$$\boldsymbol{r}_{t,\text{out}} = g(\hat{\boldsymbol{z}}_t), \tag{2}$$

where the subscripts 'in' and 'out' denote partitions of the neurons (Fig 1B). During evaluation, the held-out data from test trials $\boldsymbol{X}_{:,\text{out}}$ is compared to the rate-predictions $\boldsymbol{R}_{:,\text{out}}$ from the model using the co-smoothing metric $\mathcal{Q}$ defined as the normalised log-likelihood, given by:

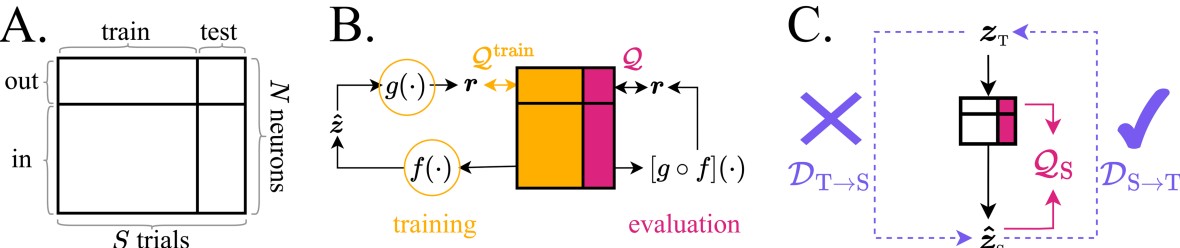

**Fig 1. Prediction framework and its relation to ground truth. A.** To evaluate a neural LVM with co-smoothing, the dataset is partitioned along the neurons and trials axes. **B.** The held-in neurons are used to infer latents $\hat{z}$, while the held-out serve as targets for evaluation. The encoder $f$ and decoder $g$ are trained jointly to maximise co-smoothing $\mathcal{Q}$. After training, the composite mapping $g \circ f$ is evaluated on the test set. **C.** We hypothesise that models with high co-smoothing may have an asymmetric relationship to the true system, ensuring that model representation contains the ground truth, but not vice-versa. We reveal this in a synthetic student(S)-teacher(T) setting by the unequal performance of regression on the states in the two directions. $\mathcal{D}_{u \to v}$ denote decoding error of model $v$ latents $\boldsymbol{z}_v$ from model $u$ latents $\boldsymbol{z}_u$.

$$Q(R_{t,n}, X_{t,n}) := \frac{1}{\mu_n \log 2}\Big(\mathcal{L}(R_{t,n}; X_{t,n}) - \mathcal{L}(\bar{r}_n; X_{t,n})\Big) \tag{3}$$

$$\mathcal{Q}^{\text{test}} := \sum_{n \in \text{held-out}} \sum_{i \in \text{test}} \sum_{t=1}^{T} Q(R_{t,n}^{(i)}, X_{t,n}^{(i)}), \tag{4}$$

where $\mathcal{L}$ is poisson log-likelihood, $\bar{r}_n = \frac{1}{TS}\sum_i \sum_t X_{t,n}^{(i)}$ is a the mean rate for channel $n$, and $\mu_n := \sum_i \sum_t X_{t,n}^{(i)}$ is the total number of spikes, following [6].

Thus, the inference of LVM parameters is performed through the optimisation:

$$f^*, g^* = \text{argmax}_{f,g} \mathcal{Q}^{\text{train}} \tag{5}$$

using $\mathcal{X}^{\text{train}}$, without access to the test trials from $\mathcal{X}^{\text{test}}$. For clarity, apart from (5), we report only $\mathcal{Q}^{\text{test}}$, omitting the superscript.

## Good co-smoothing does not guarantee correct latents

It is common to assume that being able to predict held-out parts of **X** will guarantee that the inferred latent aligns with the true one [6,14,16–28]. To test this assumption, we use a student-teacher scenario where we know the ground truth. To compare how two models ($u,v$) align, we infer the latents of both from $\mathcal{X}^{\text{test}}$, then do a regression from latents of $u$ to $v$. The regression error is denoted $\mathcal{D}_{u \to v}$ (i.e. $\mathcal{D}_{T \to S}$ for teacher to student decoding). Contrary to the above assumption, we hypothesise that good prediction guarantees that the true latents are contained within the inferred ones (low $\mathcal{D}_{S \to T}$), but not vice versa (Fig 1C). It is possible that the inferred latents possess additional features, unexplained by the true latents (high $\mathcal{D}_{T \to S}$).

We demonstrate this phenomenon in three different student-teacher scenarios: task-trained RNNs, Hidden Markov Models (HMMs) and linear gaussian state space models. We start with RNNs, as they are a standard tool to investigate computation through dynamics in neuroscience [29], and expand upon the other models in the appendix. A 128-unit RNN teacher (Methods) is trained on a 2-bit flip-flop task, inspired by working memory experiments. The network receives input pulses and has to maintain the identity of the last pulse (see Methods). The student is a sequential autoencoder, where the encoder $f$ is composed of a neural network that converts observations into an initial latent state, and another recurrent neural network that advances the latent state dynamics [29] (see Methods).

We generated a dataset of observations from this teacher, and then trained 30 students with latent-dimensionality 3–64 on the same teacher data using gradient-based methods (see Methods). Co-smoothing scores of students increased with the size of the latents, but are high for models in the range of 5-15 dimensional latents (S1 Fig). Consistent with our hypothesis, the ability to decode the teacher from the student was highly correlated to the co-smoothing score (Fig 2 top left). In contrast, the ability to decode the student from the teacher has a very different pattern. For students with low co-smoothing, this decoding is good – but meaningless. For students with high co-smoothing, there is a large variability, and little correlation to the co-smoothing score (Fig 2 top right). In this simple example, it would seem that one only needs to increase the dimensionality of the latent until co-smoothing saturates. This minimal value would satisfy both demands. This is not the case for real data, as will be shown below.

What is it about a student model, that produces good co-smoothing with the wrong latents? It's easiest to see this in a setting with discrete latents, so we first show the HMM teacher and two exemplar students – named "Good" and "Bad" (marked by green and red arrows in S3 FigAB) – and visualise their states and transitions using graphs in Fig 2. The teacher is a cycle of 4 steps. The good student contains such a cycle (orange), and the initial distribution is restricted to that cycle, rendering the other states irrelevant. In contrast, the *bad* student also contains this cycle (orange), but the initial distribution is not consistent with the cycle, leading to an extraneous branch converging to the cycle, as well as

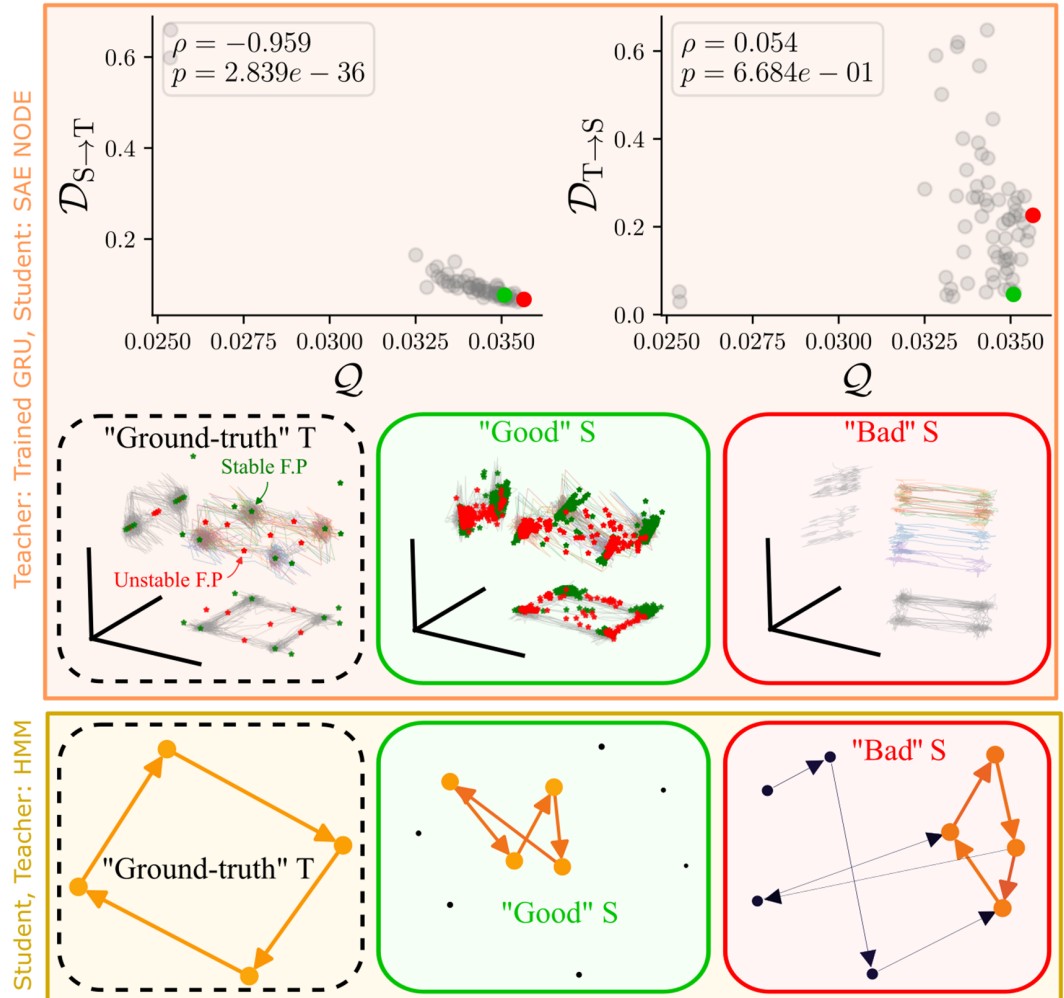

**Fig 2**. **Upper panel.** Several students, sequential autoencoders (SAE, see Methods), are trained on a dataset generated by a single teacher, a noisy GRU RNN trained on a 2-bit flip flop (2BFF, see Methods). The Student→Teacher decoding error $\mathcal{D}_{S→T}$ is low and tightly related to the co-smoothing score. The Teacher→Student decoding error $\mathcal{D}_{T→S}$ is more varied and uncorrelated to co-smoothing. A score of $\mathcal{Q} = 0$ corresponds to predicting the mean firing-rate for each neuron at all trials and time points. Green and red points are representative "Good" and "Bad" students respectively, whose latents are visualised below along-side the ground truth T. The visualisations are projections of the latents along the top three principal components of the data. The ground truth latents are characterised by 4 stable states capturing the $2^2$ memory values. This structure is captured in the "Good" student. The bad student also includes this structure in addition to an extraneous variability along the third component. **Lower panel.** The same experiment conducted with HMMs. The teacher is a nearly deterministic 4-cycle and students are fit to its noisy emissions. Dynamics in selected models are visualised. Circles represent states, and arrows represent transitions. Circle area and edge thickness reflect fraction of visitations or volume of traffic after sampling the HMM over several trials. The colours also reflect the same quantity – brighter for higher traffic. Edges with values below 0.01 are removed for clarity (S5 Fig). The teacher ($M = 4$) is a 4-cycle. Note the prominent 4-cycles (orange) present in the good student ($M = 10$), and the bad student ($M = 8$). In the good student, the extra states are seldom visited, whereas in the bad student there is significant extraneous dynamics involving these states (dark arrows).

a departure from the main cycle (both components in dark colour). Note that this does not interfere with co-smoothing, because the emission probabilities of the extra states are consistent with true states, i.e., the emission matrix conceals the extraneous dynamics. In the RNN, we see a qualitatively similar picture, with the bad students having dynamics in task-irrelevant dimensions (Fig 2 "Bad" S).

## Few-shot prediction selects better models

Because our objective is to obtain latent models that are close to the ground truth, the co-smoothing prediction scores described above are not satisfactory. Can we devise a new prediction score that will be correlated with ground truth similarity? The advantage of prediction benchmarks is that they can be optimised, and serve as a common language for the community as a whole to produce better algorithms [30].

We suggest **few-shot co-smoothing** as a complementary prediction score to co-smoothing, to be used on models with good scores on the latter. Similarly to standard co-smoothing, the functions $g$ and $f$ are trained using all trials of the training data (Fig 3A). The key difference is that a separate group of $N^{k\text{-out}}$ neurons is set aside (Table 1), and only $k$ trials of these neurons are used to estimate a mapping $g' : \hat{\mathbf{Z}}_{t,:} \mapsto \mathbf{R}_{t,k\text{-out}}$ (Fig 3B), similar to $g$ in (2). The neural LVM ($f, g, g'$) is then evaluated on both the standard co-smoothing $\mathcal{Q}$ using $g \circ f$ and the few-shot version $\mathcal{Q}^k$ using $g' \circ f$ (Fig 3C).

For small values of $k$, the $\mathcal{Q}^k$ scores can be highly variable. To reduce this variability, we repeat the procedure $s$ times on independently resampled sets of $k$ trials, producing $s$ estimates of $g'$, each with its own score $\mathcal{Q}^k$. For each student S, we then report the average score $\langle \mathcal{Q}_S^k \rangle$ across the $s$ resamples. A theoretical analysis of the choice of $k$ is given in the next section, with practical guidelines provided in S2 Fig. The number of resamples $s$ is chosen empirically to ensure high confidence in the estimated average (Methods).

To demonstrate the utility of the proposed prediction score, we return to the RNN students from Fig 2 and evaluate $\langle \mathcal{Q}_S^k \rangle$ for each. This score provides complementary information about the models, as it is uncorrelated with standard co-smoothing (Fig 4A), and it is not merely a stricter version of co-smoothing (S6 Fig). Since we are only interested in models with good co-smoothing, we restrict attention to students satisfying $\mathcal{Q}_S > \mathcal{Q}_T - 10^{-3}$. Among these students, despite their nearly identical co-smoothing scores, the $k$-shot scores $\langle \mathcal{Q}_S^k \rangle$ are strongly correlated with the ground-truth measure $\mathcal{D}_{T \to S}$ (Fig 4B). Together, these findings suggest that simultaneously maximizing $\mathcal{Q}_S$ and $\langle \mathcal{Q}_S^k \rangle$—both prediction-based

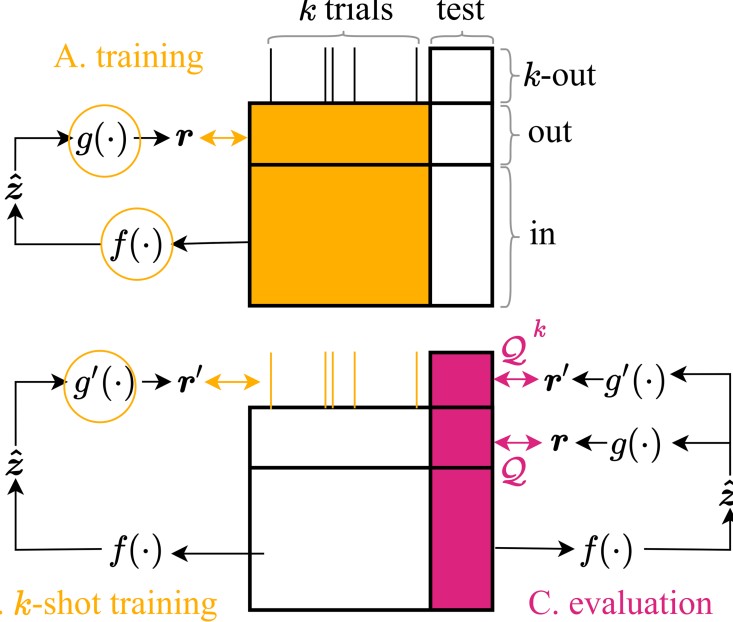

**Fig 3**. **Co-smoothing and few-shot co-smoothing; a composite evaluation framework for Neural LVMs. A.** The encoder $f$ and decoder $g$ are trained jointly using held-in and held-out neurons. **B.** A separate decoder $g'$ is trained to readout $k$-out neurons using only $k$ trials. Meanwhile, $f$ and $g$ are frozen. **C.** The neural LVM is evaluated on the test set resulting in two scores: co-smoothing $\mathcal{Q}$ and $k$-shot co-smoothing $\mathcal{Q}^k$.

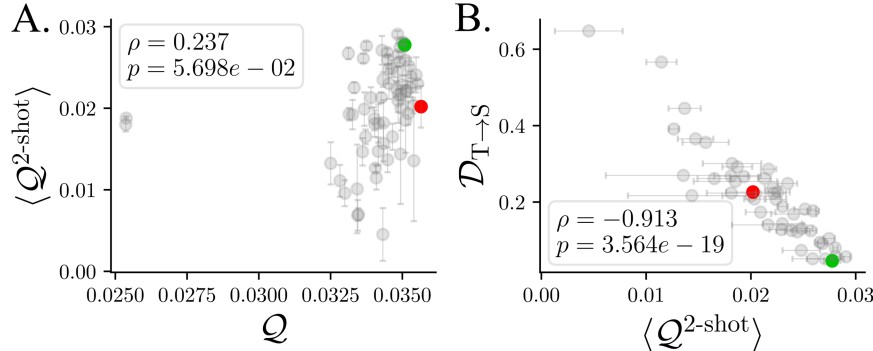

**Fig 4**. **Few-shot prediction selects better models. A.** Few-shot measures something new. Student models with high co-smoothing have highly variable 2-shot co-smoothing, which is uncorrelated to co-smoothing. Error bars reflect standard error of the mean across several few-shot regressions (see Methods). **B.** For the set of students with high co-smoothing, i.e., satisfying $\mathcal{Q} > 0.034$, 2-shot co-smoothing to held-out neurons is negatively correlated with decoding error from teacher-to-student. Green and red points represent the example "Good" and "Bad" models (Fig 2).

objectives—produces models with low $\mathcal{D}_{S \to T}$ and $\mathcal{D}_{T \to S}$, yielding a more complete measure of model similarity to the ground truth.

**Why does few-shot work?**

The example HMM and RNN students of Fig 2 can help us understand why few-shot prediction identifies good models. The students differ in that the *bad* student has more than one state corresponding to the same teacher state. Because these states provide the same output, this feature does not hurt co-smoothing. In the few-shot setting, however, the output of all states needs to be estimated using a limited amount of data. Thus the information from the same amount of observations has to be distributed across more states. We make this data efficiency argument more precise in three settings: linear regression (LR), HMMs, and binary classification prototype learning (BCPL).

In the case of LR, the teacher latent is a scalar random variable $z$ and the student latent $\hat{z}$ is a random $p$-vector, whose first coordinate is $z$ and the remaining $p$–1 coordinates are the extraneous noise:

$$\hat{z} := \begin{bmatrix} z & \underbrace{\xi_1 & \xi_2 & \cdots & \xi_{p-1}}_{\text{extraneous noise}} \end{bmatrix}^T, \tag{6}$$

where $\xi_j \sim \mathcal{N}(0, \sigma_{\text{ext}}^2)$. In other words, a single teacher state is represented by several possible student states.

Next, we model the neural-data – noisy observations of the teacher latent $x := z + \epsilon$, where $\epsilon \sim (0, \sigma_{\text{obs}}^2)$. The few-shot learning is captured by minimum-norm $k$-shot least-squares linear regression:

$$\hat{w} := \arg\min_w \left\{ \|w\|^2 : w \text{ minimises } \sum_{i=1}^{k} \|x^{(i)} - w^T \hat{z}^{(i)}\|^2 \right\}, \tag{7}$$

where $\| \cdot \|$ is the 2-norm.

The generalisation error of the few-shot learner is given by:

$$\mathcal{R}^k = \left\langle (\hat{z}^T w^* - \hat{z}^T \hat{w})^2 \right\rangle_{z, \xi_1, \dots, \xi_p, \epsilon}, \tag{8}$$

where $w^* = \begin{bmatrix} 1 & 0 & \cdots & 0 \end{bmatrix}^T$ is the true mapping.

We solve for $\langle \mathcal{R}^k \rangle$ as $k, p \to \infty$, $p/k \to \gamma \in (0, \infty)$ using the theory of [31], and demonstrate a good fit to numerical simulations at finite $p,k$ (Methods). We do similar analyses for Bernoulli HMM latents with maximum likelihood estimation of the emission parameters (Methods) and BCPL [32] (Methods).

Across the three scenarios, model performance decreases with extraneous variability (Fig 5). Crucially, this difference appears at small $k$, and vanishes as $k \to \infty$. With HMMs and BCPL this is a gradual decrease, while in LR, there is a known critical transition at $p = k$ [31,33,34].

Interestingly, the scenarios differ in the bias-variance decomposition of their performance deficits. In LR, extraneous noise leads to increased bias with identical variance (Methods, Claim 2), whereas in the HMM and BCPL, it leads to increased variance and zero bias (Methods, (28) and (52) respectively).

How does one choose the value of $k$ in practice? The intuition and theoretical results suggest that we want the smallest possible value. In real data, however, we expect many sources of noise that could make small values impractical. For instance, for low firing rates, small $k$ values can mean that some neurons will not have any spikes in $k$ trials and thus there will nothing to regress from. Our suggestion is therefore to use the smallest value of $k$ that allows robust estimation of few-shot co-smoothing. (S2 Fig) shows the effect of this choice for various datasets.

### SOTA LVMs on neural data

In previous sections, we showed that models with near perfect co-smoothing may possess latents with extraneous dynamics. We established this in a synthetic student-teacher setting with RNNs, HMMs and LGSSM models.

To show the applicability in more realistic scenarios, we consider four datasets `mc_maze_20` [35], `mc_rtt_20` [36], `dmfc_rsg_20` [37], `area2_bump_20` [38] from the Neural Latent Benchmarks suite [6] (see Methods). They consist of neural activity (spikes) recorded from various cortical regions of monkeys as they perform specific tasks. The `20` indicates that spikes were binned into 20$ms$ time bins. We trained several SpatioTemporal Neural Data Transformers (STNDTs) [39–42], that achieve near state-of-the-art (SOTA) co-smoothing on these datasets. We evaluate co-smoothing on a test set of trials and define the set of models with the best co-smoothing (see Methods and Table 1).

A key component of training modern neural network architectures such as STNDT is the random sweep of hyperparameters, a natural step in identifying an optimal model for a specific data set [19]. This process generates several candidate

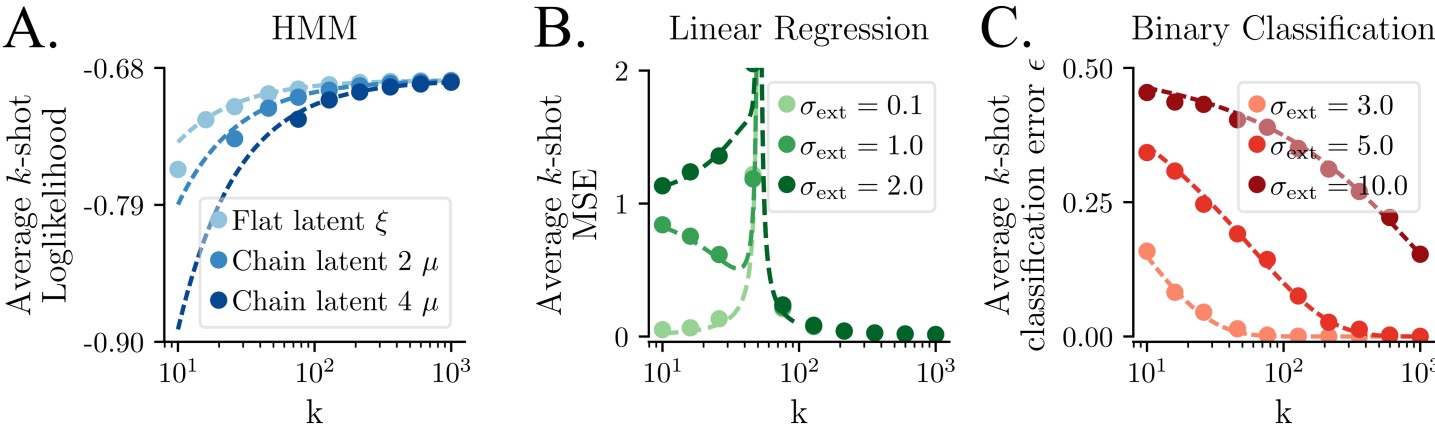

**Fig 5**. **Theoretical analysis of $k$-shot learner performance as a function of $k$ and extraneous noise $\sigma_{ext}$, in three different settings.** Points show numerical simulations and dashed lines show analytical theory. **A.** Hidden Markov Models (HMMs) (Methods), Bernoulli observations, MLE estimator. **B.** Minimum norm least squares linear regression with $\sigma_{obs} = 0.3$ and $p = 50$ (main text and Methods). **C.** binary classification, prototype learning (Methods).

solutions to the optimisation problem (5), yielding models with similar co-smoothing scores but, as we demonstrate in this section, varying amounts of extraneous dynamics.

**Two proxies for $\mathcal{D}_{\text{T} \rightarrow \text{S}}$: cycle consistency and cross-decoding.**

To reveal extraneous dynamics in the synthetic examples (RNNs, HMMs), we had access to ground truth that enabled us to directly compare the student latent to that of the teacher. With real neural data, we do not have this privilege. This limitation has been recognised in the past and a proxy was suggested [8,29,43] – *cycle consistency*. Instead of decoding the student latent from the teacher latent, cycle consistency attempts to decode the student latent $\hat{z}$ from the student's own *rate prediction* $r$. In our notation this is $\mathcal{D}_{r \rightarrow \hat{z}}$ (Fig 6A and Methods). If the student has perfect co-smoothing (see S3 Appendix), this should be equivalent to $\mathcal{D}_{\text{T} \rightarrow \text{S}}$ as it would ensure that teacher and student have the same rate-predictions $r$.

Because we cannot rely on perfect co-smoothing, we also suggest a novel metric – *cross-decoding* – where we compare the models to each other. The key idea is that all high co-smoothing models contain the teacher latent. One can then imagine that each student contains a selection of several extraneous features. The best student is the one containing the least such features, which would imply that all other students can decode its latents, while it cannot decode theirs (Fig 6B). Instead of computing $\mathcal{D}_{\text{S} \rightarrow \text{T}}$ and $\mathcal{D}_{\text{T} \rightarrow \text{S}}$ as in Fig 2, we perform decoding from latents of model $u$ to model $v$ ($\mathcal{D}_{u \rightarrow v}$) for every pair of models $u$ and $v$ using linear regression and evaluating an $R^2$ score for each mapping (see Methods). In Fig 6C the results are visualised by a $U \times U$ matrix with entries $\mathcal{D}_{u \rightarrow v}$ for all pairs of models $u$ and $v$. The ideal model $v^*$ would have no extraneous dynamics, therefore, all the other models should be able to decode its latents perfectly, i.e., $\mathcal{D}_{u \rightarrow v^*} = 0 \ \forall \ u$. Provided a large and diverse population of models only the 'pure' ground truth would satisfy this condition. To evaluate how close a model $v$ is to the ideal $v^*$ we propose a simple metric: the column average $\langle \mathcal{D}_{u \rightarrow v} \rangle_u$. This will serve as proxy for the distance to ground truth, analogous to $\mathcal{D}_{\text{T} \rightarrow \text{S}}$ in Fig 4. We validate this procedure using the RNN student-teacher setting in Fig 6D, where we show that $\langle \mathcal{D}_{u \rightarrow v} \rangle_u$ is highly correlated to the ground truth measure $\mathcal{D}_{\text{T} \rightarrow \text{S}}$. We also validate cycle-consistency $\mathcal{D}_{r \rightarrow \hat{z}}$ against $\mathcal{D}_{\text{T} \rightarrow \text{S}}$ using the RNN setting (Fig 6E). In both cases we find a high correlation between the metrics.

Having developed a proxy for the ground truth we can now correlate it with the few-shot co-smoothing $\langle Q^{k\text{-shot}} \rangle$ to held-out neurons. Following the discussion in the previous section, we choose the smallest value of $k$ that ensures no trials with zero spikes (S2 Fig). Fig 7 shows a negative correlation of $\langle Q^{k\text{-shot}} \rangle$ with both proxy measures $\mathcal{D}_{r \rightarrow \hat{z}}$ and $\langle \mathcal{D}_{u \rightarrow v} \rangle_u$ across the STNDT models in the four data sets. Moreover, regular co-smoothing $Q$ for the same models is relatively uncorrelated with these measures. As an illustration of the latents of different models, Fig 7(bottom) shows the PCA projection of latents from two STNDT models trained on `mc_maze_20`. Both have high co-smoothing scores but differ in their few-shot scores $\langle Q^{k\text{-shot}} \rangle$. We note smoother trajectories and better clustering of conditions in the model with higher $\langle Q^{k\text{-shot}} \rangle$. We also quantify the ability to decode behavior from these two models, and found the top-PCs perform better in the "Good" model (S7 Fig).

## Discussion

Latent variable models (LVMs) aim to infer the underlying latents using observations of a target system. We showed that co-smoothing, a common prediction measure of the goodness of such models, cannot discriminate between LVMs containing only the true latents and those with additional extraneous dynamics.

We propose a complementary prediction measure: few-shot co-smoothing. After training the encoder that translates data observations to latents, we use only a few ($k$) trials to train a new decoder. Using several synthetic datasets generated from trained RNNs and two other state-space architectures, we show numerically and analytically that this measure correlates with the distance of model latents to the ground truth.

We demonstrate the applicability of this measure to four datasets of monkey neural recordings with a transformer architecture [39,40] that achieves near state-of-the-art (SOTA) results on all datasets. This required developing a new proxy to

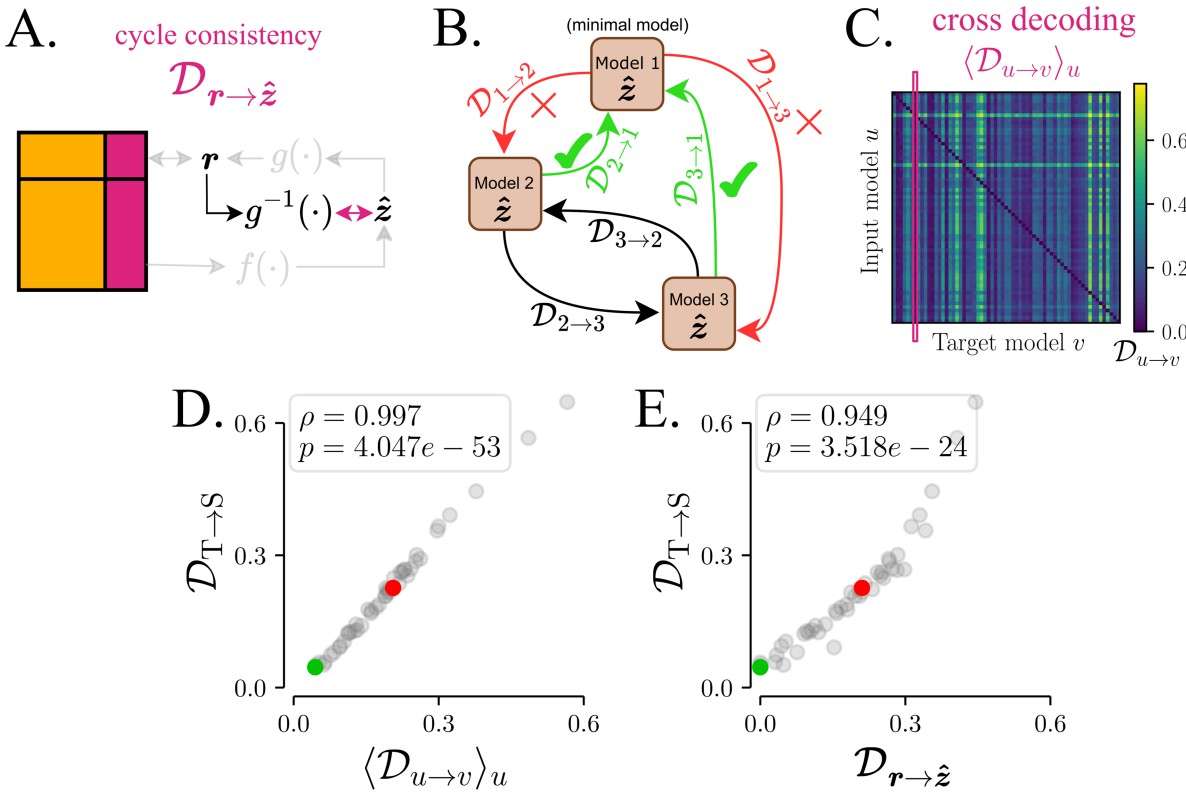

**Fig 6. Cycle consistency and cross-decoding as a proxy for distance to the ground truth in the absence of ground-truth. A** *Cycle consistency* $\mathcal{D}_{r \to \hat{z}}$ [8,29,43] involves learning a mapping $g^{-1}$ from the rates $r$ back to the latents $\hat{z}$ (see Methods). **B** The latents of each pair of models are *cross-decoded* from one another. Minimal models can be fully decoded by all models but extraneous models only by some. **C** Cross-decoding matrix for SAE NODE models trained on data from the NoisyGRU (Fig 2). **D, E** For models with high co-smoothing ($\Omega > 0.035$) the proxy metrics – cross-decoding column average $\langle \mathcal{D}_{u \to v} \rangle_u$, and cycle-consistency $\mathcal{D}_{r \to z}$ – are both highly correlated to ground truth $\mathcal{D}_{T \to S}$.

ground truth – cross-decoding. For each pair of models, we try to decode the latents of one from the latents of the other. Models with extraneous dynamics showed up as poor target latents on average, and vice versa.

Our work is related to a recent study that addresses benchmarking LVMs for neural data by developing benchmarks and metrics using only synthetic data - Computation through dynamics benchmark [29]. This study similarly tackles the issue of extraneous dynamics, primarily using ground-truth comparisons and cycle consistency. Our cross-decoding metric complements cycle consistency [8,29] as a proxy for ground truth. Cycle consistency has the advantage that it is defined on single models, compared with cross-decoding that depends on the specific population of models used. Cycle consistency has the disadvantage that it uses the rate predictions as proxies to the true dynamics. In the datasets we analyzed here, both measures provided very similar results. An interesting extension would be to use the cross-decoding metric as another method to select good models. However, its computational cost is high, as it requires training a population of models and comparing them pairwise. Additionally, it is less universal and standardised than few-shot co-smoothing, as it depends on a specific 'jury' of models.

Several works address the issue of extraneous dynamics through regularisation of dimensionality, picking the minimal dimensional or rank-constrained model that still fits the data [8,11–13]. Usually, these constraints are accompanied by poorer co-smoothing scores compared to their unconstrained competitors, and the simplicity of these constrained models often goes uncredited by standard prediction-based metrics. Classical measures like AIC [44] and BIC [45] address

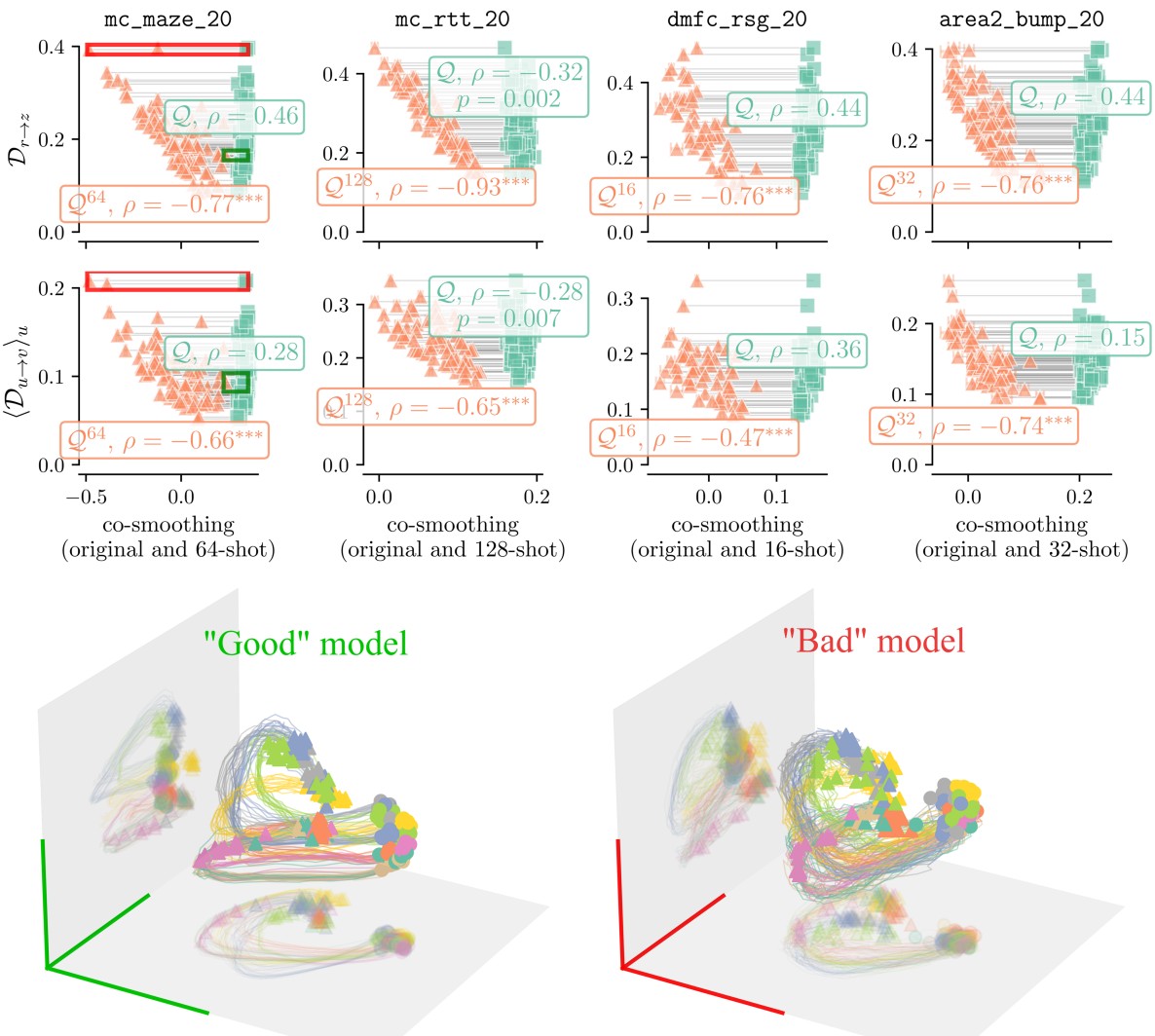

**Fig 7. Few-shot scores $\langle\mathcal{Q}^{k\text{-shot}}\rangle$ correlate with the proxies of distance to the ground truth, cycle-consistency $\mathcal{D}_{r\to z}$ and the cross-decoding column average $\langle\mathcal{D}_{u\to v}\rangle_u$.** We train several STNDT models on four neural recordings from monkeys [35–38], curated by [6] and filter for models with high co-smoothing $\mathcal{Q} > 0.8 \times \max(\mathcal{Q})$. The few-shot co-smoothing scores $\langle\mathcal{Q}^{k\text{-shot}}\rangle$ negatively correlate with the two proxies $\mathcal{D}_{r\to z}$ and $\langle\mathcal{D}_{u\to v}\rangle_u$ (orange points), while regular co-smoothing $\mathcal{Q}$ (turquoise points) does not (one-tailed p-values shown for $p < 0.05$ and *** for $p < 0.001$). Green and red arrows indicate the extreme models whose latents are visualised below. $\mathcal{Q}$ values may be compared against an EvalAI leaderboard [6]. Note that we evaluate using an offline train-test split, not the true test set used for the leaderboard scores, for which held-out neuron data is not publicly accessible. (Bottom) Principal component analysis of the latent trajectories of two STNDT models trained on `mc_maze_20` with similar co-smoothing scores but contrasting few-shot co-smoothing. The "Good" model scores $\mathcal{Q} = 0.341$, $\langle\mathcal{Q}^{64\text{-shot}}\rangle = 0.292$ and the "Bad" model $\mathcal{Q} = 0.342$, $\langle\mathcal{Q}^{64\text{-shot}}\rangle = 0.012$. The trajectories are coloured by task conditions and start at a circle and end in a triangle.

the issue of overfitting by penalising the number of parameters, but are less applicable given the success of overparameterised models [33]. We believe these approaches may not scale well to increasingly larger datasets [46], noting studies reporting that neural activity is not finite-dimensional but exhibits a scale-free distribution of variance [47,48]. Our few-shot co-smoothing metric, by contrast, does not impose dimensional constraints and instead leverages predictive performance on limited data to identify models closer to the true latent dynamics, potentially offering better scalability for complex, large-scale neural datasets. Furthermore, limiting the method to prediction offers other advantages. Prediction

PLOS Computational Biology

benchmarks are a common language for the community to optimise inference methods, without requiring access to the latents, which could be model-specific.

While the combination of student-teacher and SOTA results presents a compelling argument, we address a few limitations of our work. Regarding few-shot regression, while the Bernoulli HMM and linear regression scenarios have a closed-form solutions, the Poisson GLM regression for SOTA models is optimised iteratively and is sensitive to the L2 hyperparameter $\alpha$. In our results, we select a minimal $\alpha$ that is sufficient to stabilise optimisation.

A broader limitation concerns LVM architectures with varying decoder ($g$) parameterisations, which would in general require different few-shot learning procedures for the auxiliary decoder ($g'$). Our results show that few-shot scores are indicative of model extraneousness when comparing models with a fixed decoder architecture. In our SOTA experiments, we use a conventional linear–exponential–Poisson decoder. However, when comparing models with substantially different decoder architectures—such as multi-layer nonlinear decoders [11] or linear–Gaussian emission models [27,49]—differences in few-shot performance may reflect strengths or weaknesses of the few-shot learning procedure in the respective setting, rather than differences in the extraneousness of the inferred latents.

Overall, our work advances latent dynamics inference in general and prediction frameworks in particular. By exposing a failure mode of standard prediction metrics, we guide the design of inference algorithms that account for this issue. Furthermore, the few-shot co-smoothing metric can be incorporated into existing benchmarks, helping the community build models that are closer to the desired goal of uncovering latent dynamics in the brain.

## Methods

### Glossary

**Latent variable model (LVM)** ($f$ and $g$) : A function mapping neural time-series data to an inferred latent space ($f$). The latents can then be used to predict held-out data ($g$).

**Smoothing** : mapping a sequence of observations $\boldsymbol{X}_{1:T}$ to a sequence of inferred latents $\hat{\boldsymbol{Z}}_{1:T}$. It is often formalised as a conditional probability $p(\hat{\boldsymbol{Z}}_{1:T}|\boldsymbol{X}_{1:T})$.

**Extraneous dynamics** : the notion that inferred latent variables may contain features and temporal structure not present in the true system from which the data was observed.

**Co-smoothing** ($\mathcal{Q}$) : A metric evaluating LVMs by their ability to predict the activity of held-out neurons $\boldsymbol{X}_{1:T,\text{out}}$ provided held-in neural activity $\boldsymbol{X}_{1:T,\text{in}}$ over a window of time. The two sets of neurons are typically random subsets from a single population.

**Few-shot co-smoothing** ($\mathcal{Q}^{k\text{-shot}}$) : A variant of co-smoothing in which the mapping from latents to held-out neurons ($g'$) is learned from a small number of trials.

**State-of-the-art (SOTA)** : the best performing method or model current available in the field. This is usually based on a specific benchmark, i.e., a dataset and associated evaluation metric. In active fields the SOTA is constantly improving.

**Cycle consistency** ($\mathcal{D}_{r\to\hat{z}}$) : a measure of *extraneousness* of model latents as compared to their rate predictions. Computed by learning and evaluating the inverse mapping from rate predictions to latents.

**Cross-decoding** ($\mathcal{D}_{u\to v}$) : another measure of model *extraneousness*. It is evaluated on a population of models trained on the same dataset. It involves regressing from one model latents to another model, for all pairs in the population. A scalar measure is the obtained for each model: the cross-decoding column mean $\langle\mathcal{D}_{u\to v}\rangle_u$. It reflects the average 'decodability' of a model, by all the other models.

### Student-teacher Recurrent Neural Networks (RNN)

Both teacher and student are based on an adapted version of [29]. In the following, we provide a brief description.

**Teacher**

We train a noisy 64 Gated Recurrent Unit (NoisyGRU) RNN [50], on a 2-bit flip flop 2BFF task [3], implemented by [29]. The GRU RNN follows standard dynamics, which we repeat here using the typical notation of GRUs. This notation is not consistent with the Results section, and we explain the relation below.

$$\mathbf{h}_0 = \mu + \eta; \quad \eta \sim \mathcal{N}(0, 0.05) \tag{9}$$

$$\mathbf{z}_t = \sigma(\mathbf{W}_z \mathbf{x}_t + \mathbf{U}_z \mathbf{h}_{t-1} + \mathbf{b}_z) \tag{10}$$

$$\mathbf{r}_t = \sigma(\mathbf{W}_r \mathbf{x}_t + \mathbf{U}_r \mathbf{h}_{t-1} + \mathbf{b}_r) \tag{11}$$

$$\tilde{\mathbf{h}}_t = \tanh(\mathbf{W}_h \mathbf{x}_t + \mathbf{U}_h(\mathbf{r}_t \odot \mathbf{h}_{t-1}) + \mathbf{b}_h + \xi_t); \quad \xi_t \sim \mathcal{N}(0, 0.01) \tag{12}$$

$$\mathbf{h}_t = (1 - \mathbf{z}_t) \odot \mathbf{h}_{t-1} + \mathbf{z}_t \odot \tilde{\mathbf{h}}_t, \tag{13}$$

where $\eta$, $\mathbf{W}_z$, $\mathbf{U}_z$, $\mathbf{b}_z$, $\mathbf{W}_r$, $\mathbf{U}_r$, $\mathbf{b}_r$, $\mathbf{W}_h$, $\mathbf{U}_h$, $\mathbf{b}_h$ are trainable parameters. The latent used in the Results section (*z*) is the hidden unit activity *h*. After model training, the NoisyGRU units are subsampled, centered, normalised, and rectified to give synthetic neural firing rates - which are *r* of the Results section. These firing rates are used to define a stochastic Poisson process to generate the synthetic neural data.

**Students**

The student models are sequential autoencoders (SAEs) consisting of a bidirectional GRU that predicts the initial latent state, a Neural ODE (NODE) that evolves the latent dynamics (together these form the encoder, *f*, under our notation), and a linear readout layer mapping the latent states to the data (the decoder, *g*). We train several randomly initialised models with a range of latent dimensionalities (3, 5, 8 : 16, 32, 64). Models are trained to minimise a Poisson negative loglikelihood reconstruction loss, using the Adam [51] optimiser.

**Student-teacher Hidden Markov Models (HMMs)**

We choose both student and teacher to be discrete-space, discrete-time Hidden Markov Models (HMMs). As a teacher model, they simulate two important properties of neural time-series data: its dynamical nature and its stochasticity. As a student model, they are perhaps the simplest LVM for time-series, yet they are expressive enough to capture real neural dynamics ($\mathcal{Q}$ of 0.29 for HMMs vs. 0.24 for GPFA and 0.35 for LFADS, on `mc_maze_20`). The HMM has a state space $z \in \{1, 2, \dots, M\}$, and produces observations (emissions in HMM notation) along neurons $\boldsymbol{X}$, with a state transition matrix $\boldsymbol{A}$, emission model $\boldsymbol{B}$ and initial state distribution $\pi$. More explicitly:

$$\begin{aligned} A_{m,l} &= p(z_{t+1} = l | z_t = m) && \forall\, m, l \\ B_{m,n} &= p(x_{n,t} = 1 | z_t = m) && \forall\, m, n \\ \pi_m &= p(z_0 = m) && \forall\, m \end{aligned} \tag{14}$$

The same HMM can serve two roles: a) data-generation by sampling from (14) and b) inference of the latents from data on a trial-by-trial basis:

$$\xi_{t,m}^{(i)} = f_m((\boldsymbol{X}_{:,\text{in}})^{(i)}) = p(z_t^{(i)} = m | (\boldsymbol{X}_{:,\text{in}})^{(i)}), \tag{15}$$

i.e., *smoothing*, computed exactly with the forward-backward algorithm [52]. Note that although *z* is the latent state of the HMM, we use its posterior probability mass function $\xi_t$ as the relevant intermediate representation because it reflects a richer representation of the knowledge about the latent state than a single discrete state estimate. To make predictions of

the rates of held-out neurons for co-smoothing we compute:

$$R_{n,t}^{(i)} = g_n(\boldsymbol{\xi}_t^{(i)}) = \sum_m B_{m,n} \xi_{t,m}^{(i)}. \tag{16}$$

As a teacher, we constructed a 4-state model of a noisy chain $A_{m,l} \propto \mathbb{I}[l = (m + 1) \mod M] + \epsilon$, with $\epsilon = 1e - 2$, $\pi = \frac{1}{M}$, and $B_{m,n} \sim \text{Unif}(0, 1)$ sampled once and frozen (Fig 2, left). We generated a dataset of observations from this teacher (see Table 1).

For each student, we evaluate $\langle \mathcal{Q}_S^k \rangle$. This involves estimating the bernoulli emission parameters $\hat{\boldsymbol{B}}_{m,k\text{-out}}$, given the latents $\xi_{t,m}^{(i)}$ using (26) and then generating rate predictions for the $k$-out neurons using (16).

## HMM training

HMMs are traditionally trained with expectation maximisation, but they can also be trained using gradient-based methods. We focus here on the latter as these are used ubiquitously and apply to a wide range of architectures. We use an existing implementation of HMMs with differentiable parameters: dynamax [53] – a library of differentiable state-space models built with jax.

We seek HMM parameters $\theta := (A, B^{[\text{in,out}]}, \pi)$ that minimise the negative log-likelihood loss, $L$ of the held-in and held-out neurons in the train trials:

$$L(\theta; \mathcal{X}_{[\text{in,out}]}^{\text{train}}) = -\log p(\mathcal{X}_{[\text{in,out}]}^{\text{train}}; \theta) \tag{17}$$

$$= \sum_{i \in \text{train}} -\log p\left(\left(X_{1:T,[\text{in,out}]}\right)^{(i)}; \theta\right) \tag{18}$$

To find the minimum we do full-batch gradient descent on $L$, using dynamax together with the Adam optimiser [51] .

## Decoding across HMM latents

Consider two HMMs $u$ and $v$, of sizes $M(u)$ and $M(v)$, both candidate models of a dataset $\mathcal{X}$. Following (15), each HMM can be used to infer latents from the data, defining encoder mappings $f^u$ and $f^v$. These map a single trial $i$ of the data $(\boldsymbol{X}_{:,\text{in}})^{(i)} \in \mathcal{X}$ to $(\boldsymbol{\xi}_t^{(i)})_u$ and $(\boldsymbol{\xi}_t^{(i)})_v$.

Since HMM latents are probability mass functions, we do not do use linear regression to learn the mappings across model latents. Instead we perform a multinomial regression from $(\boldsymbol{\xi}_t^{(i)})_u$ to $(\boldsymbol{\xi}_t^{(i)})_v$.

$$\boldsymbol{p}_t^{(i)} = h\left(\left(\boldsymbol{\xi}_t^{(i)}\right)_u\right) \tag{19}$$

$$h(\xi) = \sigma(W\boldsymbol{\xi} + \boldsymbol{b}) \tag{20}$$

where $W \in \mathbb{R}^{M(v) \times M(u)}$, $\boldsymbol{b} \in \mathbb{R}^{M(v)}$ and $\sigma$ is the softmax. During training we sample states from the target PMFs $(z_t^{(i)})_v \sim (\boldsymbol{\xi}_t^{(i)})_v$ thus arriving at a more well-known problem scenario: classification of $M(v)$-classes. We optimise $W$ and $\boldsymbol{b}$ to minimise a cross-entropy loss to the target $(\hat{z}_t^{(i)})_v$ using the `fit()` method of `sklearn.linear_model.LogisticRegression`.

We define decoding error, as the average Kullback-Leibler divergence $D_{KL}$ between target and predicted distributions:

$$\mathcal{D}_{u \to v} := \frac{1}{S^{\text{test}} T} \sum_{i \in \text{test}} \sum_{t=1}^{T} D_{KL}\left(\boldsymbol{p}_t^{(i)}, (\boldsymbol{\xi}_t^{(i)})_v\right) \tag{21}$$

where $D_{KL}$ is implemented with `scipy.special.rel_entr`.

In section  and Fig 1, the data $X$ is sampled from a single teacher HMM, T, and we evaluate $\mathcal{D}_{T \to S}$ and $\mathcal{D}_{S \to T}$ for each student notated simply as S.

## Analysis of LVMs without access to ground truth

We denote the set of high co-smoothing models as those satisfying $\mathcal{Q} > 0.034$ for Fig 4 and $\mathcal{Q} > 0.8 \times \mathcal{Q}_{\text{best model}}$ in Fig 7, $\mathcal{F} := \{(f_u, g_u)\}_{u=1}^{U}$, the the encoders and decoders respectively. Note that STNDT is a deep neural network given by the composition $g \circ f$, and the choice of intermediate layer whose activity is deemed the 'latent' $\mathbf{Z}$ is arbitrary. Here we consider $g$ the last 'read-out' layer and $f$ to represent all the layers up-to $g$.

### Few-shot co-smoothing

To perform few-shot co-smoothing, we learn $g'$, which takes the same form as $g$, a Poisson Generalised Linear Model (GLM) for each held-out neuron. We use `sklearn.linear_model.PoissonRegressor`, which has a hyperparameter `alpha`, the amount of l2 regularisation. For the results in the main text, $\langle \mathcal{Q}^{k\text{-shot}} \rangle$ in Fig 7, we select $\alpha = 10^{-3}$. We partition the training data into several random subsets of $k$ trials and train an independently initialised GLM on each subset. Each GLM is then evaluated on a fixed test set of trials (Fig 3), yielding a score for each subset. We report the mean over $\lfloor 5 \times S^{\text{train}}/k \rfloor$ such repetitions, $\langle \mathcal{Q}^{k\text{-shot}} \rangle$, along with the standard error of the mean (error bars in Fig 4, Fig 7). Scores are more variable at small $k$, so we need more repetitions to better estimate the average score. To implement this in a standarised way, we incorporate this chunking of data into several subsets in the `nlb_tools` library (Data and code availability). This way we ensure that all models are trained and tested on identical subsets. We report the compute-time for few-shot co-smoothing in S2 Appendix.

### Cross-decoding

We perform a cross-decoding from the latents of model $u$, $(\mathbf{Z}_{t,:})_u$, to those of model $v$, $(\mathbf{Z}_{t,:})_v$, for every pair of models $u$ and $v$ using a linear mapping $h(\mathbf{z}) := \mathbf{W}\mathbf{z} + \mathbf{b}$ implemented with `sklearn.linear_model.LinearRegression`:

$$\left( \hat{\mathbf{z}}_{t,:}^{(i)} \right)_v = h_{u \to v}\left( \left( \mathbf{Z}_{t,:}^{(i)} \right)_u \right) \tag{22}$$

minimising a mean squared error loss. We then evaluate a $R^2$ score (`sklearn.metrics.r2_score`) of the predictions, $(\hat{\mathbf{Z}})_v$, and the target, $(\mathbf{Z})_v$, for each mapping. We define the decoding error $\mathcal{D}_{u \to v} := 1 - (R^2)_{u \to v}$. The results are accumulated into a $U \times U$ matrix (see Fig 6).

### Cycle consistency

We evaluate cycle-consistency [8,29] for a model $u$ also using a linear mapping from its rate predictions $\mathbf{R}$ back to its latents $\hat{\mathbf{Z}}$ implemented with `sklearn.linear_model.LinearRegression`:

$$\left( \hat{\mathbf{z}}_{t,:}^{(i)} \right)_u = h_{r \to \hat{z}}\left( \left( \mathbf{R}_{t,\text{out}}^{(i)} \right)_u \right), \tag{23}$$

again minimising a squared error loss. As in cross-decoding we evaluate $R^2$ score (`sklearn.metrics.r2_score`) and the decoding error $\mathcal{D}_{r \to \hat{z}} := 1 - (R^2)_{r \to \hat{z}}$ (Fig 6A).

## Summary of Neural Latent Benchmark (NLB) datasets

Here are brief descriptions of the datasets used in this study. All datasets were collected from macaque monkeys performing sensorimotor or cognitive tasks. More comprehensive details can be found in the Neural Latents Benchmark paper [6].

`mc_maze` [35] Motor cortex recordings during a delayed reaching task where monkeys navigated around virtual barriers to reach visually cued targets. The task involved 108 unique maze configurations, with several repeated trials for each one, thus serving as a "neuroscience MNIST". We choose this dataset to visualise the latents in Fig 7.

`mc_rtt` [36] Motor cortex recordings during naturalistic, continuous reaching toward randomly appearing targets without imposed delays. The task lacks trial structure and includes highly variable movements, emphasizing the need for modeling unpredictable inputs and non-autonomous dynamics.

`dmfc_rsg` [37] Recordings from dorsomedial frontal cortex during a time-interval reproduction task, where monkeys estimated and reproduced time intervals between visual cues using eye or hand movements. The task involves internal timing, variable priors, and mixed sensory-motor demands.

`area2_bump` [38] Somatosensory cortex recordings during a visually guided reach task in which unexpected mechanical bumps to the limb occurred in half of the trials. The task probes proprioceptive feedback processing and requires modeling input-driven neural responses.

## Dimensions of datasets

We analyse several datasets in this work. Three synthetic datasets generated by an RNN, HMM (Methods, Fig 2) and LGSMM (S4 Fig) and the four datasets from the Neural Latent Benchmarks (NLB) suite [6,35–38]. In Table 1, we summaries the dimensions of all these datasets. To evaluate $k$-shot on the existing SOTA methods while maintaining the NLB evaluations, we conserved the *forward-prediction* aspect. During model training, models output rate predictions for $T^{\text{fp}}$ future time bins in each trial, i.e., (1) and (2) are evaluated for $1 \leq t \leq T^{\text{fp}}$ while input remains as $\boldsymbol{X}_{1:T,\text{in}}$. Although we do not discuss the forward-prediction metric in our work, we note that the SOTA models receive gradients from this portion of the data.

In all the NLB datasets as well as the RNN dataset we reuse held-out neurons as $k$-out neurons. We do this to preserve NLB evaluation metrics on the SOTA models, as opposed to re-partitioning the dataset resulting in different scores from previous works. This way existing co-smoothing scores are preserved and $k$-shot co-smoothing scores can be directly compared to the original co-smoothing scores. The downside is that we are not testing the few-shot on 'novel' neurons. Our numerical results (Fig 7) show that our concept still applies.

## Theoretical analysis of few shot learning in HMMs.

Consider a student-teacher scenario as in section . We let $T = 2$ and use a stationary teacher $z_1^{(i)} = z_2^{(i)}$. Now consider two examples of inferred students. To ensure a fair comparison, we use two latent states for both students. In the *good* student, $\xi$, these two states statistically do not depend on time, and therefore it does not have extraneous dynamics. In contrast, the *bad* student, $\mu$, uses one state for the first time step, and the other for the second time step. A particular

Table 1. **Dimensions of real and synthetic datasets.** Number of train and test trials $S^{\text{train}}$, $S^{\text{test}}$, time-bins per trial for co-smoothing $T$, and forward-prediction $T^{\text{fp}}$, held-in, held-out and $k$-out neurons $N^{\text{in}}$, $N^{\text{out}}$, $N^{k\text{-out}}$. [†]In all the NLB [6] datasets as well the RNN dataset we use the same set of neurons for $N^{\text{out}}$ and $N^{k\text{-out}}$.

| DATASET | $S^{\text{train}}$ | $S^{\text{test}}$ | $T$ | $T^{\text{fp}}$ | $N^{\text{in}}$ | $N^{\text{out}}$ | $N^{k\text{-out}}$ |
|---|---|---|---|---|---|---|---|
| SYNTHETIC NOISY GRU RNN (METHODS) [29] | 800 | 200 | 500 | – | 50 | 10 | 10[†] |
| SYNTHETIC HMM (METHODS) | 2000 | 100 | 10 | – | 20 | 50 | 50 |
| SYNTHETIC LGSSM (S4 FIG) | 20 | 500 | 10 | – | 5 | 30 | 30 |
| `mc_maze_20` [35] | 1721 | 574 | 35 | 10 | 137 | 45 | 45[†] |
| `mc_rtt_20` [36] | 810 | 270 | 30 | 10 | 98 | 32 | 32[†] |
| `dmfc_rsg_20` [37] | 748 | 258 | 75 | 10 | 40 | 14 | 14[†] |
| `area2_bump_20` [38] | 272 | 92 | 30 | 10 | 49 | 16 | 16[†] |

example of such students is:

$$\xi_t = \begin{bmatrix} 0.5 & 0.5 \end{bmatrix}^T \quad t \in \{1, 2\} \tag{24}$$

$$\mu_{t=1} = \begin{bmatrix} 1 & 0 \end{bmatrix}^T \qquad \mu_{t=2} = \begin{bmatrix} 0 & 1 \end{bmatrix}^T \tag{25}$$

where each vector corresponds to the two states, and we only consider two time steps.

We can now evaluate the maximum likelihood estimator of the emission matrix from $k$ trials for both students. In the case of bernoulli HMMs the maximum likelihood estimate of $g'$ given a fixed $f$ and $k$ trials has a closed form:

$$\hat{B}_{m,n} = \frac{\sum_{i \in k\text{-shot trials}} \sum_{t=1}^{T} \mathbb{I}[X_{t,n}^{(i)} = 1] \xi_{t,m}^{(i)}}{\sum_{i' \in k\text{-shot trials}} \sum_{t'=1}^{T} \xi_{t',m}^{(i')}} \qquad \forall \ 1 \leq m \leq M \text{ and } n \in k\text{-out neurons} \tag{26}$$

We consider a single neuron, and thus omit $n$, reducing the estimates to:

$$\begin{aligned} \hat{B}_1(\xi) &= \frac{0.5(C_1 + C_2)}{0.5kT} & \hat{B}_1(\mu) &= \frac{C_1}{k} \\ \hat{B}_2(\xi) &= \frac{0.5(C_1 + C_2)}{0.5kT} & \hat{B}_2(\mu) &= \frac{C_2}{k} \end{aligned} \tag{27}$$

where $C_t$ is the number of times $x = 1$ at time $t$ in $k$ trials. We see that $C_t$ is a sum of $k$ i.i.d. Bernoulli random variables (RVs) with the teacher parameter $B^*$, for both $t = 1, 2$.

Thus, $\hat{B}_m(\xi)$ and $\hat{B}_m(\mu)$ are scaled binomial RVs with the following statistics:

$$\begin{aligned} \mathbb{E}\hat{B}_1(\xi) = \mathbb{E}\hat{B}_2(\xi) = B^* & \qquad \mathbb{E}\hat{B}_1(\mu) = \mathbb{E}\hat{B}_2(\mu) = B^* \\ \text{Cov}\left[\hat{\boldsymbol{B}}(\xi)\right] = \frac{1}{2k} B^*(1 - B^*) \begin{bmatrix} 1 & 1 \\ 1 & 1 \end{bmatrix} & \qquad \text{Cov}\left[\hat{\boldsymbol{B}}(\mu)\right] = \frac{1}{k} B^*(1 - B^*) \begin{bmatrix} 1 & 0 \\ 0 & 1 \end{bmatrix} \end{aligned} \tag{28}$$

The test loss is given by $L(\hat{B}) = \mathbb{E}\frac{1}{T} \sum_t \log p(X_t^{(i)}; \hat{B}) = \frac{1}{T} \sum_t B^* \log(R_t) + (1 - B^*) \log(1 - R_t)$. For $\xi$, $R_t = 0.5(\hat{B}_1 + \hat{B}_2)$ for both values of $t$, and for $\mu$, $R_1 = \hat{B}_1$ and $R_2 = \hat{B}_2$. Ultimately,

$$L_\xi(\hat{\boldsymbol{B}}(\xi)) = \frac{1}{T} \sum_t B^* \log\left(0.5(\hat{B}_1 + \hat{B}_2)\right) + (1 - B^*) \log\left(1 - 0.5(\hat{B}_1 + \hat{B}_2)\right) \tag{29}$$

$$L_\mu(\hat{\boldsymbol{B}}(\mu)) = \frac{1}{T} \sum_t B^* \log\left(\hat{B}_t\right) + (1 - B^*) \log\left(1 - \hat{B}_t\right) \tag{30}$$

To see how these variations affect the test loglikelihood $L$ of the few-shot regression on average, we do a taylor expansion around $B^*$, recognising that the function is maximised at $B^*$, so $\left.\frac{\partial L}{\partial B}\right|_{B^*} = 0$.

$$\mathbb{E}_{\hat{B}_k} L(\hat{B}_k) = \mathbb{E}_{\hat{B}_k}\left[ L(B_\infty) + \frac{1}{2}(\hat{B}_k - B^*)^T \left.\frac{\partial^2 L}{\partial B^2}\right|_{B^*} (\hat{B}_k - B^*) + \dots \right] \tag{31}$$

$$\approx L(B^*) + \mathbb{E}_{\hat{B}_k} \frac{1}{2}(\hat{B}_k - B^*)^T \left.\frac{\partial^2 L}{\partial B^2}\right|_{B^*} (\hat{B}_k - B^*) \tag{32}$$

$$= L(B^*) + \underbrace{\frac{1}{2}(\mathbb{E}\hat{B}_k - B^*)^T \left.\frac{\partial^2 L}{\partial B^2}\right|_{B^*} (\mathbb{E}\hat{B}_k - B^*)}_{\text{bias}} + \underbrace{\frac{1}{2}\text{Tr}\left[ \text{Cov}(\hat{B}_k) \left.\frac{\partial^2 L}{\partial B^2}\right|_{B^*} \right]}_{\text{variance}} \tag{33}$$

We see that this second order truncation of the loglikelihood is decomposed into a bias and a variance term. We recognise that the bias term goes to zero because we know the estimator is unbiased ((28)). To compute the variance term, we compute the hessians which differ for the two models:

$$\frac{\partial^2 L_\xi}{\partial B^2}\bigg|_{B^*} = -\frac{\eta}{4}\begin{bmatrix} 1 & 1 \\ 1 & 1 \end{bmatrix}, \qquad\qquad \frac{\partial^2 L_\mu}{\partial B^2}\bigg|_{B^*} = -\frac{\eta}{2}\begin{bmatrix} 1 & 0 \\ 0 & 1 \end{bmatrix}, \qquad (34)$$

where $\eta = \frac{1}{B^*(1-B^*)}$.

Incorporating these hessians into (33), we obtain:

$$\mathbb{E}_{\hat{B}_k} L_\xi\left(\hat{B}_k(\xi)\right) \approx L(B^*) - \frac{1}{8k}\,\text{Tr}\begin{bmatrix} 2 & 2 \\ 2 & 2 \end{bmatrix} = L(B^*) - \frac{1}{2k}, \qquad (35)$$

$$\mathbb{E}_{\hat{B}_k} L_\mu\left(\hat{B}_k(\mu)\right) \approx L(B^*) - \frac{1}{2k}\,\text{Tr}\begin{bmatrix} 1 & 0 \\ 0 & 1 \end{bmatrix} = L(B^*) - \frac{1}{k}. \qquad (36)$$

Fig 5A shows these analytical results against the left hand side of (35) and (36) evaluated numerically.

**Theoretical analysis of ridgeless least squares regression with extraneous noise.**

Teacher latents $z_i^* \sim \mathcal{N}(0, 1)$ generate observations $x_i$:

$$x_i = z_i^* + \epsilon_i, \qquad (37)$$

where $\epsilon_i \sim \mathcal{N}(0, \sigma_{\text{obs}}^2)$ is observation noise.

In this setup there is no time index: we consider only a single sample index $i$.

We consider candidate student latents, $\mathbf{z} \in \mathbb{R}^p$, that *contain* the teacher along with extraneous noise, i.e:

$$\mathbf{z}_i := \begin{bmatrix} z_i^* & \xi_i \end{bmatrix}^T, \qquad (38)$$

where $\xi_i \sim \mathcal{N}(0, \sigma_{\text{ext}}^2 \mathbf{I}_{p-1})$ is a vector of i.i.d. extraneous noise, and $\mathbf{I}_{p-1}$ is the $(p-1) \times (p-1)$ identity matrix.

We study the minimum $l_2$ norm least squares regression estimator on $k$ training samples:

$$\hat{\mathbf{w}} = \arg\min\left\{\|w\|_2 \;:\; w \text{ minimises } \sum_{i=1}^{k} \|x_i - \mathbf{w}^T \mathbf{z}_i\|_2^2\right\}. \qquad (39)$$

with the regression weights $\mathbf{w} \in \mathbb{R}^p$. More succinctly, $\mathbf{z}_i \sim \mathcal{N}(0, \Sigma)$, where $\Sigma = \text{diag}([1, \sigma_{\text{ext}}^2, \dots, \sigma_{\text{ext}}^2])$.

Note that, by construction, the true mapping is:

$$\mathbf{w}^* = \begin{bmatrix} 1 & 0 & \dots & 0 \end{bmatrix}^T. \qquad (40)$$

Test loss or *risk* is a mean squared error:

$$R(\hat{\mathbf{w}}; \mathbf{w}^*) = \mathbb{E}_{\mathbf{z}_0}\left(\mathbf{z}_0^T \mathbf{w}^* - \mathbf{z}_0^T \hat{\mathbf{w}}\right)^2, \qquad (41)$$

given a test sample $z_0$. The error can be decomposed as:

$$R(\hat{w}; w^*) = \underbrace{\|\mathbb{E}(\hat{w}) - w^*\|_\Sigma^2}_{\text{bias, } B} + \underbrace{\text{Tr}\left[\text{Cov}(\hat{w})\Sigma\right]}_{\text{variance, } V},\tag{42}$$

The scenario described above is a special case of [31]. What follows is a direct application of their theory, which studies the risk $R$, in the limit $k, p \to \infty$ such that $p/k \to \gamma \in (0, \infty)$, to our setting. The alignment of the theory with numerical simulations is demonstrated in Fig 5B.

**Claim 1.** $\gamma < 1$, i.e., the underparameterised case $k > p$.
$B = 0$ and the risk is just variance and is given by:

$$\lim_{k,p\to\infty \text{ and } p/k\to\gamma} R(\hat{w}; w^*) = \sigma_{\text{obs}}^2 \frac{\gamma}{1-\gamma},\tag{43}$$

with no dependence on $\sigma_{\text{ext}}$.

*Proof*: This is a direct restatement of Proposition 2 in [31]. $\square$

**Claim 2.** $\gamma > 1$, i.e., the overparameterised case $k < p$.
The following is true as $k, p \to \infty$ and $p/k \to \gamma$:

$$\lim_{k,p\to\infty \text{ and } p/k\to\gamma} B = \frac{\gamma(\gamma - 1)}{\left(\gamma - 1 + \frac{1}{\sigma_{\text{ext}}^2}\right)^2}\tag{44}$$

$$\lim_{k,p\to\infty \text{ and } p/k\to\gamma} V = \sigma_{\text{obs}}^2 \frac{\gamma}{\gamma - 1}\tag{45}$$

*Proof*: For the non-isotropic case [31] define the following distributions based on the eigendecomposition of $\Sigma$.

$$d\widehat{H}(s) = \frac{1}{p}\delta(s - 1) + \frac{p - 1}{p}\delta(s - \sigma_{\text{ext}}^2)\tag{46}$$

$$d\widehat{G}(s) = \delta(s - 1)\tag{47}$$

In the limit $p \to \infty$ we take $d\widehat{H}(s) \approx \delta(s - \sigma_{\text{ext}}^2)$. This greatly simplifies calculations and nevertheless provide a good fit for numerical results with finite $k$ and $p$. We solve for $c_0(\gamma, \widehat{H})$ using equation 12 in [31].

$$\gamma c_0 = \frac{1}{(\gamma - 1)\sigma_{\text{ext}}^2}\tag{48}$$

We then compute the limiting values of $B$ and $V$:

$$B = \|w^*\|^2(1 + \gamma c_0\sigma_{\text{ext}}^2)\frac{1}{(1 + \gamma c_0)^2}\tag{49}$$

$$V = \sigma_{\text{obs}}^2 \gamma c_0 \sigma_{\text{ext}}^2.\tag{50}$$

Substituting $\gamma c_0$ completes the proof. $\square$

The extraneous noise, $\sigma_{\text{ext}}$, influences the risk of ridgeless regression only in the regime $k < p$, and its effect is confined to the bias term, leaving the variance unaffected. In contrast, observation noise contributes exclusively to the variance term. Consequently, the dependence of the risk on $\sigma_{\text{ext}}$ persists even in the absence of observation noise, i.e., when $\sigma_{\text{obs}} = 0$.

Fig 5B presents the theoretical predictions alongside the empirical average $k$-shot performance of minimum-norm least-squares regression, computed numerically using the function `numpy.linalg.lstsq`.

## Theoretical analysis of prototype learning for binary classification with extraneous noise

Teacher latents are distributed as $p(z_i^*) = \frac{1}{2}\delta(z_i^* - \frac{1}{\sqrt{2}}) + \frac{1}{2}\delta(z_i^* + \frac{1}{\sqrt{2}})$, that is either $\frac{1}{\sqrt{2}}$ or $-\frac{1}{\sqrt{2}}$ with probability $\frac{1}{2}$, representing two classes $a$ and $b$ respectively.

We consider candidate student latents, $\boldsymbol{z} \in \mathbb{R}^{2M+1}$, that *contain* the teacher along with extraneous noise, i.e:

$$\boldsymbol{z}_i := \begin{cases} \begin{bmatrix} z_i^* & \xi_i & \boldsymbol{0} \end{bmatrix}^T & \text{if } z_i^* = 1 \\ \begin{bmatrix} z_i^* & \boldsymbol{0} & \xi_i \end{bmatrix}^T & \text{if } z_i^* = -1 \end{cases} \tag{51}$$

where $\xi_i \sim \mathcal{N}(0, \sigma_{\text{ext}}^2 I_M)$ is a $M$-vector of i.i.d. extraneous noise, and $I_M$ is the $M \times M$ identity matrix and $\boldsymbol{0} \in \mathbb{R}^M$.

We consider the prototype learner $\boldsymbol{w} = \bar{\boldsymbol{z}}_a - \bar{\boldsymbol{z}}_b$, $b = \frac{1}{2}(\bar{\boldsymbol{z}}_a + \bar{\boldsymbol{z}}_b)$, where $\bar{\boldsymbol{z}}_a$ and $\bar{\boldsymbol{z}}_b$ are the sample means of $k$ latents from class $a$ and $k$ latents from class $b$ respectively. The classification rule is given by the sign of $\boldsymbol{w}^T\boldsymbol{x} - b$: classifying the input $\boldsymbol{x}$ as $a$ if positive and $b$ otherwise.

This setting is a special case of [54]. They provide a theoretical prediction for average few-shot classification error rate for class $a$, $\epsilon_a$, given by $\epsilon_a = H(SNR)$ where $H(x) = \frac{1}{\sqrt{2\pi}}\int_x^\infty dt \exp(-t^2/2)$ is a monotonously decreasing function.

$$SNR_a = \frac{1}{2}\frac{\|\Delta\boldsymbol{x}_0\|^2 + (R_b^2 R_a^2 - 1)/k}{\sqrt{D_a^{-1}/k + \|\Delta\boldsymbol{x}_0^T U_b\|^2/k + \|\Delta\boldsymbol{x}_0^T U_a\|^2}}. \tag{52}$$

$\Delta\boldsymbol{z} = \boldsymbol{z}_a - \boldsymbol{z}_b$ the difference of the population centroids of the two classes.
In our case this reduces to:

$$SNR \approx \frac{\sqrt{Mk}}{\sigma_{\text{ext}}^2} \tag{53}$$

To obtain this we note radii of manifold $a$ is $\begin{bmatrix} 0 & \sigma_{\text{ext}} & \dots & \sigma_{\text{ext}} & 0 & \dots & 0 \end{bmatrix}$ with an average radius $R = R_a = R_b = \frac{M}{(2M+1)}\sigma_{\text{ext}}^2$ and participation ratio $D_a = \left(\sum_i (R_a^i)^2\right)^2 / \sum_i (R_a^i)^4 = M$.

We substitute $\|\Delta\boldsymbol{x}_0\|^2 = \frac{1}{R^2} = \frac{2M+1}{M\sigma_{\text{ext}}^2} \approx \frac{2}{\sigma_{\text{ext}}^2}$.

The bias term $(R_b^2 R_a^2 - 1)/k$ is zero since $R_a = R_b$.

The $\Delta\boldsymbol{x}_0^T U_a$ and $\Delta\boldsymbol{x}_0^T U_b$ terms are both zero.

The participation ratio $D_a = M$. Our construction is symmetric in that $SNR_a = SNR_b$.

The classification error, $\epsilon$, decreases monotonically with the number of samples $k$, tending to zero as $k \to \infty$ for all finite values of $\sigma_{\text{ext}}$. In contrast, $\epsilon$ increases monotonically with extraneous noise $\sigma_{\text{ext}}$, deviating significantly from zero once $\sigma_{\text{ext}}^2 \approx \sqrt{Mk}$.

Fig 5C presents the numerically computed error in comparison with the theoretical prediction given in (53).

## Data and code availability

The experiments done in this work are largely based on code repositories from previous works. The following repositories were used or developed in this work:

- https://github.com/KabirDabholkar/ComputationThroughDynamicsBenchmark.git – Code from Versteeg et al. [29], which we used directly for training and analysis of RNNs and NODE SAEs.
- https://github.com/KabirDabholkar/hmm_analysis - Training and analysis of HMMs, implemented in `dynamax` [53]
- https://github.com/KabirDabholkar/ssm_analysis - Training and analysis of LGSSMs, implemented in `dynamax` [53]
- https://github.com/KabirDabholkar/nlb_tools_fewshot – A fork of the Neural Latents Benchmark repository by Pei et al. [6], used for evaluation of state-of-the-art models (includes co-smoothing, few-shot co-smoothing, cycle-consistency, and cross-decoding).
- https://github.com/KabirDabholkar/STNDT_fewshot - Training STNDT models [6,21,40–42]

## Supporting information

**S1 Fig. Student-Teacher RNNs: co-smoothing as a function of model size.** Finding the correct model is not just about tuning the latent size hyperparameter. NODE SAE students over a range of sizes (5-15) achieve high co-smoothing on the same 64-unit noisy GRU performing 3BFF teacher (Methods).
(TIFF)

**S2 Fig. How to choose $k$ for your dataset?** Our theoretical analysis in "Why does few-shot work?" reveals that extraneous models are best discriminated when the shot number, $k$, is small. So how small can we go? In the case of sparse data like neural spike counts we may obtain $k$-trial subsets in which some neurons are silent. In this scenario the few-shot decoder $g'$ receives no signal for those neurons. To avoid this pathological scenario, for each dataset, we pick the smallest possible $k$ that ensures that the probability of encountering silent neurons in a $k$-trial subset is safely near zero. This must be computed for each dataset independently since some datasets are more sparse than others. We compute the frequency of such silences for different $k$, for each NLB [6] dataset, and show the values of $k$ (dashed lines) chosen for the analysis in the main text.
(TIFF)

**S1 Appendix. Decoding across HMM latents: fitting and evaluation.**
(PDF)

**S3 Fig. Good co-smoothing does not guarantee correct latents in Hidden Markov Models (HMMs).** In the main text, we show how good prediction of held-out neural activity, i.e., *co-smoothing*, does not guarantee a match between model and true latents. We did this in the student-teacher setting of RNNs and NODE SAEs (Fig 2). Here we replicate the results in HMMs (see Methods). Similar to Fig 2, several students HMMs are trained on a dataset generated by a single teacher HMM, a noisy 4-cycle. The Student→Teacher decoding error $\mathcal{D}_{S \to T}$ is low and tightly related to the co-smoothing score. The Teacher→Student decoding error $\mathcal{D}_{T \to S}$ is more varied and uncorrelated to co-smoothing. The arrows mark the "Good" and "Bad" transition matrices shown in the Fig 2 (lower).
(TIFF)

**S4 Fig. Student-teacher results in Linear Gaussian State Space Models.** We demonstrate that our results are not unique to the RNN or HMM settings by simulating another simple scenario: linear gaussian state space models (LGSSM), i.e., Kalman Smoothing.

The model is defined by parameters $(\boldsymbol{\mu}_0, \boldsymbol{\Sigma}_0, \boldsymbol{F}, \boldsymbol{G}, \boldsymbol{H}, \boldsymbol{R})$. A major difference to HMMs is that the latent states $\boldsymbol{z} \in \mathbb{R}^M$ are continuous. They follow the dynamics given by:

$$\boldsymbol{z}_0 \sim \mathcal{N}(\boldsymbol{\mu}_0, \boldsymbol{\Sigma}_0) \tag{54}$$

$$\boldsymbol{z}_t \sim \mathcal{N}(\boldsymbol{F}\boldsymbol{z}_{t-1} + \boldsymbol{b}, \boldsymbol{G}) \tag{55}$$

$$\boldsymbol{x}_t \sim \mathcal{N}(\boldsymbol{H}\boldsymbol{z}_t + \boldsymbol{c}, \boldsymbol{R}) \tag{56}$$

Given these dynamics, the latents $\boldsymbol{z}$ can be inferred from observations $\boldsymbol{x}$ using Kalman smoothing, analogous to (15). Here we use the jax based dynamax implementation.

We use a teacher LGSSM with $M = 4$, with parameters chosen randomly (using the dynamax defaults) and then fixed. Student LGSSMs are also initialised randomly and optimised with Adam [51] to minimise negative loglikelihood on the training data (see the dataset dimensions section for dimensions of the synthetic data set). $\mathcal{D}_{S\rightarrow T}$ and $\mathcal{D}_{T\rightarrow S}$ is computed with linear regression (`sklearn.linear_model.LinearRegression`) and predictions are evaluated against the target using $R^2$ (`sklearn.metrics.r2_score`). We define $\mathcal{D}_{u\rightarrow v} := 1 - (R^2)_{u\rightarrow v}$. Few-shot regression from $\boldsymbol{z}$ to $\boldsymbol{x}^{k-\text{out}}$ is also performed using linear regression.

In line with our results with RNNs and HMMs (Fig 2 and Fig 4), we show that among the models with high test log-likelihood (>–55), $\mathcal{D}_{S\rightarrow T}$, but not $\mathcal{D}_{T\rightarrow S}$, is highly correlated to test loglikelihood, while $\mathcal{D}_{T\rightarrow S}$ shows a close relationship to Average 10 shot MSE error. For these Linear Gaussian State Space Models, we report loglikelihood instead of co-smoothing, and $k$-shot MSE instead of $k$-shot co-smoothing, demonstrating the same pattern of results across different model classes.
(TIFF)

**S5 Fig. HMM network visualisations.** In the main text Fig 2 we visualised the teacher and two student HMMs as graphs of fractional traffic volume on states and transitions. For clarity we dropped the low probability edges with values lower than 0.01. We also show the same models with all the edges visualised, including the low probability transitions that were omitted in the main text figure for clarity.
(TIFF)

**S6 Fig. Few-shot co-smoothing is not simply hard co-smoothing (variations of HMM student-teacher experiments).** Few-shot co-smoothing is a more difficult metric than standard co-smoothing. Thus, it might seem that any increase in the difficulty of will yield similar results. To show this is not the case, we use standard co-smoothing with fewer held-in neurons. The score is lower (because it's more difficult), but does not discriminate models.

We demonstrate this through two variations of HMM student-teacher experiments. In the first variation, we increase the number of held out neurons from $N^{\text{out}} = 50$ to $N^{\text{out}} = 100$, making the co-smoothing problem harder. The top three panels show: (1) decoder student-teacher original simple, (2) decoder teacher-student original simple (same as main text Fig 1CD), and (3) decoder teacher-student 6-shot best (same as main text Fig 4B). In the second variation, we decrease the number of held-in and held-out neurons to $N^{\text{in}} = 5$, $N^{\text{out}} = 5$, $N^{k-\text{out}} = 50$, further increasing difficulty. The bottom three panels show the same three decoder configurations as the top row. While the score does decrease because the problem is harder, co-smoothing is still not indicative of good models while few-shot co-smoothing remains discriminative.
(TIFF)

**S2 Appendix. Time cost of computing few-shot co-smoothing.**
(PDF)

**S7 Fig. Classifying task variables from latents in models with contrasting few-shot performance.** In main text Fig 7(lower panel), we compare two STNDT models trained on `mc_maze_20` that perform identically under standard

co-smoothing but diverge under 64-shot co-smoothing. Projecting their latents onto the top two principal components reveals differences in trajectory smoothness and task-condition separation. Quantitatively, the "Bad" model exhibits higher latent dimensionality, as reflected by the slower growth of variance explained across PCs (left panel), and yields poorer binary classification of maze barrier presence—especially when using only the top two principal components (right panel). (TIFF)

**S3 Appendix. Illustrative example of the difference between cycle consistency and cross-decoding.** (PDF)

## Financial disclosure

This work was supported by the Israel Science Foundation (grant No. 1442/21 to OB) and Human Frontiers Science Program (HFSP) research grant (RGP0017/2021 to OB). The funders had no role in study design, data collection and analysis, decision to publish, or preparation of the manuscript.

## Author contributions

**Conceptualization:** Kabir V. Dabholkar, Omri Barak.

**Formal analysis:** Kabir V. Dabholkar, Omri Barak.

**Funding acquisition:** Omri Barak.

**Investigation:** Kabir V. Dabholkar.

**Methodology:** Kabir V. Dabholkar, Omri Barak.

**Resources:** Omri Barak.

**Software:** Kabir V. Dabholkar.

**Supervision:** Omri Barak.

**Validation:** Kabir V. Dabholkar, Omri Barak.

**Visualization:** Kabir V. Dabholkar, Omri Barak.

**Writing – original draft:** Kabir V. Dabholkar, Omri Barak.

**Writing – review & editing:** Kabir V. Dabholkar, Omri Barak.

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
