## [Decision Letter · Decision Letter 0]

16 Apr 2025

PCOMPBIOL-D-25-00336

When predict can also explain: few-shot prediction to select better neural latents

PLOS Computational Biology

Dear Dr. Dabholkar,

Thank you for submitting your manuscript to PLOS Computational Biology. After careful consideration, we feel that it has merit but does not fully meet PLOS Computational Biology's publication criteria as it currently stands. As you will see in the attached comments, one of the reviewer has raised serious concerns on unclear method descriptions, overuse of toy examples, lack of real data validation and statistical rigor, and formatting issues. We feel substantial revisions are necessary to address these concerns. Therefore, we invite you to submit a revised version of the manuscript that addresses the points raised during the review process.

Please submit your revised manuscript within 60 days Jun 16 2025 11:59PM. If you will need more time than this to complete your revisions, please reply to this message or contact the journal office at ploscompbiol@plos.org. Please include the following items when submitting your revised manuscript:

We look forward to receiving your revised manuscript.

Kind regards,

Yuanning Li

Academic Editor

PLOS Computational Biology

Hugues Berry

Section Editor

PLOS Computational Biology

**Journal Requirements:**

At this stage, the following Authors/Authors require contributions: Kabir Vinay Dabholkar, and Omri Barak. Please ensure that the full contributions of each author are acknowledged in the "Add/Edit/Remove Authors" section of our submission form.

3) Your manuscript is missing the following sections: Results, and Methods.  Please ensure all required sections are present and in the correct order. Make sure section heading levels are clearly indicated in the manuscript text, and limit sub-sections to 3 heading levels. An outline of the required sections can be consulted in our submission guidelines here:

5) Please upload a copy of Figures 4C, and D which you refer to in your text on page 8. Or, if the subfigures are no longer to be included as part of the submission please remove all reference to them within the text.

6) Please ensure that all Figure files have corresponding citations and legends within the manuscript. Currently, Figure 5 in your submission file inventory does not have an in-text citation. Please include the in-text citation of the figure.

7) We notice that your supplementary Figures, Tables, and information are included in the manuscript file. Please remove them and upload them with the file type 'Supporting Information'. Please ensure that each Supporting Information file has a legend listed in the manuscript after the references list. Please cite and label the supplementary tables and figures as “S1 Table” and “S2 Table,” "S1 Figure", S2 Figure" and so forth.

8)  Thank you for stating "Code for the HMM simulations is available at https://github.com/KabirDabholkar/hmm_analysis." This link reaches a 404 error page. Please amend this to a working link.

9) Please amend your detailed Financial Disclosure statement. This is published with the article. It must therefore be completed in full sentences and contain the exact wording you wish to be published.

10) Please ensure that the funders and grant numbers match between the Financial Disclosure field and the Funding Information tab in your submission form. Note that the funders must be provided in the same order in both places as well. Currently, the order of the funders is different in both places.

11) Please provide a completed 'Competing Interests' statement, including any COIs declared by your co-authors. If you have no competing interests to declare, please state "The authors have declared that no competing interests exist". 

**Reviewers' comments:**

Reviewer's Responses to Questions

Reviewer #1: This manuscript introduces a novel and interesting few-shot prediction metric to enhance the alignment between inferred latent variables and the true latent. Such a tool holds significant potential for modern neuroscience research, particularly in the context of large-scale population recordings. Nevertheless, the paper lacks sufficient results to support their conclusion and the presentation of the work is kind of messy.

1. When the authors challenge the common assumption on the effectiveness of co-smoothing, they can provide concrete examples in real neural data, instead of using a simplified HMM as a toy example.

2. line 95-96, "It is common to assume that being able to predict held-out parts of X will guarantee that the inferred latent aligns with the true one"

I don't think this is a common assumption. Although co-smoothing is commonly used for benchmarking LVMs, their purpose is to demonstrate their inferred latents recover the true latent signals in some way (e.g., rotation, permutation, linear combinations etc.).

Actually, this is consistent with your statment in line 101 "we hypothesize that good prediction guarantees that the true latents are contained within the inferred ones".

3. Over-parameterization usually achieves better prediction scores. By regularizing the model to have pasimonious parameters, we may achieve good alignment between the inferred latents and the true latents. This can be verified in Fig 1E, small number of latent yields smaller D_{T->S} values. I think the authors should also comment on their few-shot approach vs model selection approach (using AIC/BIC or other regularizers described in line 18-20). One is using limited data, while the other is using limited model parameters.

4. The organization of the manuscript needs major revision. The manuscript put too much space to discuss the toy example HMM and fail to validate its applicability in real LVM on neural data, whose latent are continuous values, instead of discrete states. Most people will agree 'high prediction score not necessarily yields good alignment with the true latents', thus there is no need to use 1 page to verify this with an HMM model. Simply using a LFADS or STNDT results to show the diversity of inferred latent dynamics suffices. Or you can use simulated data to provide ground truth latent.

5. In the discussion of 'why does few-shot work', you can just describe the more insightful comments. The current version is too lengthy and obscure. You can also move them to supplementary materials. I think the manuscript should include more examples of neural data to validate the performance of this few-shot prediction score.

6. The few-shot approach is very similar to bagging in ensemble learning, with an aim of reducing variance of the model estimation. In bagging, a random sample of data in a training set is selected with replacement. After generating several data samples, these weak models (g' in your work) are then trained independently. Proably you can comment your methods by connecting it with bagging.

7. The few-shot prediction approach, which is the main contribution of the manuscript, is not clearly described.

a. Line 152, you used N^{k-out}. What's the typical value for N^{in}, N^{out}, N^{k_out}?

b. The caption of Fig 3B says that 'f and g are frozen'. If so, how does Q^k help with the model training? Are you only using few-shot prediction for model evaluation?

b. If the few-shot prediction is simply used for evaluation, the metric is only used for comparing the inferred models. What if all models fail to align with the true latent?

8. The format of the manuscripts needs huge changes. It looks like a rash change from conference submission. The authors should follow the PLOS journal format and organize their manuscript according to journal standards. For example, the supporting information has subsections with a prefix of A-Z, and different sections are like randomly compiled contents without inherent connections.

9. The time cost in computing few-shot prediction metric should be presented or discussed.

10. What if the data has no trial structure? For example, LFADS uses single trial to infer the latents. Maybe you have different definitions of trials?

11. Only uses mc_maze_20 for real-world experiments. Need more realistic evidence, and maybe some neuroscientific insights about how few-shot decoding can better recover latent dynamics.

12. No statistical representation of performance comparison, but only descriptive words and figures. Need hypothesis testing to justify your assertion about data distributions (un-correlated, negatively correlated, etc).

13. The authors realized using only HMMs for synthetic data is not enough, but just discussed in the appendices and provided additional results with LGSSMs. Why not integrate with the main text in the first place?

Minor:

- Fig 2 is a bit confusing.

- Good student on the right but mentioned first in the figure caption.

- What is "edge width"? Do you mean edge value or edge weight?

- There shouldn't be "invisible edges" in a graph. Maybe use lower alpha value or other colors.

- Line 201, typo, "the good student"

- L223-L225 the paragraph is just one sentence, which could be OK in some scenarios. But it looks more like the manuscript is not well organized.

- Fig 5 is not mentioned in your main text, instead you referred to Figure 19, which has duplicated contents with Fig 5.

- In your writing, you should consistently use only one of Fig , Fig. And Figure

- The real-world dataset needs a demonstration to address a non-expert audience:

- What is the experiment setting? Why should we model it with LVMs?

- How to interpret the latent variables underlying the dataset? And what does "extraneous dynamics" mean in this scenario?

- Without background information, a non-expert reader might not understand your experiments or your results, such as the "trajectories" in Fig 5 and 19, thus doubting the whole work.

- Some important notions (smoothing, co-smoothing, few-shot co-smoothing, cross-decoding) have similar and non-intuitive names, maybe list a name table in the main text or appendix, and use abbreviations in the main text.

- Some typos ("a the" in line 89, "LGSMM" in line 478, missing section index in line 184).

Reviewer #2: This manuscript presents a new metric for evaluating latent variable models used in neuroscience, the few-shot co-smoothing score. When combined with standard co-smoothing, the new metric can identify models that fit the observed data well but that have unnecessarily complicated latent states (where the unneeded complexity is hidden by the observation model). In other words, the metric can help to identify models that have more vs less parsimonious representations of the latent dynamics. It is extremely common in modern neuroscience to fit latent state models and interpret the inferred latents in scientific terms -- so it is a huge problem that inferred latent states can be unnecessarily complicated, even when a model is a good fit to data. Hence the current manuscript is a timely and valuable contribution to the literature. It is also well-written, convincing, and thorough.

My only substantive comment is that, as the authors demonstrate, the few-shot relationship to ground truth is very sensitive to the choice of the number of few-shot neurons k, and the appropriate choice depends on the model class. So I think it is important to discuss the choice of k in the main text rather than only discussing it in the appendix.

I include some minor additional comments below.

Minor comments:

- Equation 1: I think it would be clearer to use "Z hat" here, because f is estimating the true unknown latent state

- Page 5 line 118: It would be good to justify your use of xi rather than z hat, e.g. "we use its posterior probability mass function as the relevant intermediate representation because it reflects a richer representation of the knowledge about the latent state than a single discrete state estimate" or "... because it captures the degree of belief in a given latent state rather than just the most likely discrete state" or "... because the true latent state z is unknown, and xi completely summarizes the current knowledge of it"

- Figure 4: I believe the color still represents M in this figure -- if so, please include M in the legend (like in 1D,E). Same comment for all similar figures.

- Equations 9, 10: It wasn't obvious to me right away that xi and mu were the (posterior) probabilities of the latent states at time 1, 2. It would be good to say it explicitly.

- Equation 11: missing Bhat_1(xi) subscript

- Page 8 line 197: "we see that" -- you aren't showing the bias/variance properties here, so you should instead refer the reader to the appendix.

- Page 9 line 211: Acronym SOTA used without definition ("state of the art" is used on page 2 line 29)

- Page 9 line 236: Missing reference "as in Section we"

- Page 12 line 286: Typo "arguement"

- Page 12 line 294: Wording "may be thus can evaluated"

**Have the authors made all data and (if applicable) computational code underlying the findings in their manuscript fully available?**

Reviewer #1: Yes

Reviewer #2: Yes

PLOS authors have the option to publish the peer review history of their article (what does this mean?). If published, this will include your full peer review and any attached files.

Reviewer #1: No

Reviewer #2: **Yes: **Emily P Stephen

**Figure resubmission:**
---

## [Decision Letter · Decision Letter 1]

18 Sep 2025

PCOMPBIOL-D-25-00336R1

When predict can also explain: few-shot prediction to select better neural latents

PLOS Computational Biology

Dear Dr. Dabholkar,

Thank you for submitting your manuscript to PLOS Computational Biology. After careful consideration, we feel that it has merit but does not fully meet PLOS Computational Biology's publication criteria as it currently stands. One of the reviewers raised additional concerns and suggestions that should be addressed. Therefore, we invite you to submit a revised version of the manuscript that addresses the points raised during the review process.

Please submit your revised manuscript within 60 days Nov 18 2025 11:59PM. If you will need more time than this to complete your revisions, please reply to this message or contact the journal office at ploscompbiol@plos.org. Please include the following items when submitting your revised manuscript:

We look forward to receiving your revised manuscript.

Kind regards,

Yuanning Li

Academic Editor

PLOS Computational Biology

Hugues Berry

Section Editor

PLOS Computational Biology

**Journal Requirements:**

1) We note that your Manuscript files are duplicated on your submission. Please remove any unnecessary or old files from your revision, and make sure that only those relevant to the current version of the manuscript are included.

2) Your manuscript is missing the following section: Results.  Please ensure all required sections are present and in the correct order. Make sure section heading levels are clearly indicated in the manuscript text, and limit sub-sections to 3 heading levels. An outline of the required sections can be consulted in our submission guidelines here:

3) We notice that your supplementary information (Appendices) is included in the manuscript file. Please remove them and upload them with the file type 'Supporting Information'. Please ensure that each Supporting Information file has a legend listed in the manuscript after the references list.

4) Please amend your detailed Financial Disclosure statement. This is published with the article. It must therefore be completed in full sentences and contain the exact wording you wish to be published.

5) Please ensure that the funders and grant numbers match between the Financial Disclosure field and the Funding Information tab in your submission form. Note that the funders must be provided in the same order in both places as well. Currently, the order of the grants is different in both places.

**Reviewers' comments:**

Reviewer's Responses to Questions

Reviewer #2: The revisions address my issues with the first submission, and I approve of the additional changes.

Minor comments:

- In the section "Why does few-shot work?", you present the linear regression case first without saying that's what you're doing. That is, on p7 line 142, you introduce three models (LR, HMM, prototype). The next paragraph would be clearer if you started it with "For the linear regression case..." or similar.

- In Figure 5, it took me a second to notice that in Panel A higher is better (likelihood), while in Panels B and C lower is better (error). So I didn't get right away that all three were showing the same trend with respect to extraneous noise. It would be helpful just to state it explicitly in the text and/or caption.

- On p8 lines 156-158, you compare the methods in terms of their bias/variance decompositions. I assume you are referring to the analysis in the Methods sections "Theoretical analyis of...". If so, please refer the reader to these methods section (again).

Reviewer #3: This manuscript highlights the problem of extraneous/spurious dynamics in latent variable models and introduces two evaluation approaches for identifying it: few-shot co-smoothing and cross-decoding. This problem is important and hampers interpretation and scientific conclusions drawn from these models, so suitable evaluation frameworks are a timely and valuable contribution. After the first revision, the manuscript is much clearer and more well-motivated. However, I do still have a few questions/comments about the work:

1. Given that 1) few-shot co-smoothing is still used in conjunction with standard co-smoothing, and 2) cycle consistency and cross-decoding also indicate presence of spurious dynamics (but not co-smoothing quality), what does few-shot co-smoothing exactly offer that the combination of co-smoothing and e.g., cycle consistency does not? Can few-shot co-smoothing be sufficient alone, without also evaluating standard co-smoothing? Why is it important to specifically have a prediction-based metric for parsimony of latents?

2. It is stated in the text that cycle consistency relies on the models having “perfect co-smoothing” or having “rate predictions [that are] perfect proxies of the true dynamics,” while cross-decoding does not. However, it is also stated in the text that cross-decoding also relies on the assumption that “high co-smoothing models contain the teacher latent.” How is this assumption different from that of cycle consistency?

3. Though linear-exponential-poisson readouts are most common, some LVMs do not have this emissions model (e.g., linear-softplus, MLP). I assume this makes (linear) cross-decoding not directly applicable, and I wonder if few-shot co-smoothing scores are comparable across readout models. I assume the few-shot generalization behavior would vary, especially for a higher parameter count, neural network-based readout as in ODIN (Versteeg et al. 2023), so I would appreciate some empirical exploration and/or discussion of this limitation (if I am correct in assuming that it is a limitation).

4. Though it is maybe obvious, I think the two-bit flip flop is potentially a good example to briefly discuss the consequence of spurious dynamics on accurate interpretation of the system. Are the red and green stars in Fig 2 unstable and stable fixed points? Do the spurious dynamics in the “bad” model lead to incorrect fixed point topology (as computed in Maheswaranathan et al. 2019, for example)?

5. Though maybe only directly applicable to models with strictly linear emissions models, I would appreciate some discussion of Procrustes-style metrics from Alex Williams and others, which also penalize spurious dynamics without needing two separate metrics.

6. I appreciate the qualitative difference in smoothness and separation of latents in Fig 7. Can you perform any quantitative evaluations that support this point? For example, can conditions be more accurately classified from initial conditions of the “good” model?

7. Minor typo: disucssion (line 214)

**Have the authors made all data and (if applicable) computational code underlying the findings in their manuscript fully available?**

Reviewer #2: Yes

Reviewer #3: Yes

PLOS authors have the option to publish the peer review history of their article (what does this mean?). If published, this will include your full peer review and any attached files.

Reviewer #2: **Yes: **Emily P Stephen

Reviewer #3: No

**Figure resubmission:**
---

## [Decision Letter · Decision Letter 2]

26 Nov 2025

Dear Mr Dabholkar,

We are pleased to inform you that your manuscript 'When predict can also explain: few-shot prediction to select better neural latents' has been provisionally accepted for publication in PLOS Computational Biology.

Before your manuscript can be formally accepted you will need to complete some formatting changes, which you will receive in a follow up email. A member of our team will be in touch with a set of requests. Please also consider addressing the final comments from the reviewer regarding more discussion in the text on the limitations of the proposed methods.

Best regards,

Yuanning Li

Academic Editor

PLOS Computational Biology

Hugues Berry

Section Editor

PLOS Computational Biology

Reviewer's Responses to Questions

**Comments to the Authors:**

Reviewer #3: The authors have largely addressed all of my comments and misunderstandings. I only have one remaining minor comment, which is that I would appreciate a bit more discussion in the text on the limitations of the methods proposed here for comparing models with different readout/emissions models. I think the current manuscript convincingly shows that the proposed metrics are effective for model selection across models of the same architecture, but I think it remains unclear how to compare raw few-shot co-smoothing values across architectures, especially when they have different readout models, which is essential if the metric is to be used widely for benchmarking.

There are more exceptions to the conventional linear-exp-Poisson readout than just ODIN [1]. Old-school methods like GPFA [2] and some of their extensions like (m)DLAG [3] still (unfortunately) use a linear-Gaussian emissions model, and others like SLDS often use linear-softplus (for example, in the SLDS NLB baseline). There are also methods incorporating spike history [4] and binomial/negative-binomial count models [5]. Most critically, I think there is growing interest in modelling neural dynamics on nonlinear manifolds, which has maybe most prominently led to CEBRA [6] (not really an LVM of course) but also e.g., LVMs with tuning-curve-based readout models [7,8,9].

None of these methods are really state-of-the-art in current benchmarking settings, so I don’t think this is a concerning limitation of few-shot co-smoothing and I don’t think you need to cite every single one of these, but I think it would be better to more clearly acknowledge this potential limitation.

[1] Versteeg, C., Sedler, A. R., McCart, J. D., & Pandarinath, C. (2023). Expressive dynamics models with nonlinear injective readouts enable reliable recovery of latent features from neural activity. ArXiv, arXiv-2309.

[2] Yu, B. M., Cunningham, J. P., Santhanam, G., Ryu, S., Shenoy, K. V., & Sahani, M. (2008). Gaussian-process factor analysis for low-dimensional single-trial analysis of neural population activity. Advances in neural information processing systems, 21.

[3] Gokcen, E., Jasper, A., Xu, A., Kohn, A., Machens, C. K., & Yu, B. M. (2023). Uncovering motifs of concurrent signaling across multiple neuronal populations. Advances in Neural Information Processing Systems, 36, 34711-34722.

[4] Zhao, Y., & Park, I. M. (2017). Variational latent gaussian process for recovering single-trial dynamics from population spike trains. Neural computation, 29(5), 1293-1316.

[5] Keeley, S., Zoltowski, D., Yu, Y., Smith, S., & Pillow, J. (2020). Efficient non-conjugate Gaussian process factor models for spike count data using polynomial approximations. In International conference on machine learning (pp. 5177-5186). PMLR.

[6] Schneider, S., Lee, J. H., & Mathis, M. W. (2023). Learnable latent embeddings for joint behavioural and neural analysis. Nature, 617(7960), 360-368.

[7] Wu, A., Roy, N. A., Keeley, S., & Pillow, J. W. (2017). Gaussian process based nonlinear latent structure discovery in multivariate spike train data. Advances in neural information processing systems, 30.

[8] Jensen, K., Kao, T. C., Tripodi, M., & Hennequin, G. (2020). Manifold GPLVMs for discovering non-Euclidean latent structure in neural data. Advances in Neural Information Processing Systems, 33, 22580-22592.

[9] Genkin, M., Shenoy, K. V., Chandrasekaran, C., & Engel, T. A. (2025). The dynamics and geometry of choice in the premotor cortex. Nature, 1-9.

**Have the authors made all data and (if applicable) computational code underlying the findings in their manuscript fully available?**

Reviewer #3: Yes

PLOS authors have the option to publish the peer review history of their article (what does this mean?). If published, this will include your full peer review and any attached files.

Reviewer #3: No

---

## [Editor Report · Acceptance letter]

PCOMPBIOL-D-25-00336R2

When predict can also explain: few-shot prediction to select better neural latents

Dear Dr Dabholkar,

I am pleased to inform you that your manuscript has been formally accepted for publication in PLOS Computational Biology. Your manuscript is now with our production department and you will be notified of the publication date in due course.

With kind regards,

Anita Estes
